# Fast Controlled Generation from Language Models with Adaptive Weighted Rejection Sampling

**Benjamin Lipkin**[*1] **Benjamin LeBrun**[*5] **Jacob Hoover Vigly**[1] **João Loula**[1]
**David R. MacIver**[8] **Li Du**[6] **Jason Eisner**[6] **Ryan Cotterell**[2] **Vikash Mansinghka**[1]
**Timothy J. O'Donnell**[‡3,4,5] **Alexander K. Lew**[‡1,7] **Tim Vieira**[‡2]
[1]MIT  [2]ETH Zürich  [3]McGill  [4]Canada CIFAR AI Chair  [5]Mila
[6]Johns Hopkins  [7]Yale  [8]CHI FRO

## Abstract

The dominant approach to generating from language models subject to some constraint is *locally constrained decoding* (LCD), incrementally sampling tokens at each time step such that the constraint is never violated. Typically, this is achieved through *token masking*: looping over the vocabulary and excluding non-conforming tokens. There are two important problems with this approach. (i) Evaluating the constraint on every token can be prohibitively expensive—LM vocabularies often exceed $100,000$ tokens. (ii) LCD can distort the global distribution over strings, sampling tokens based only on local information, even if they lead down dead-end paths. This work introduces a new algorithm that addresses both these problems. First, to avoid evaluating a constraint on the full vocabulary at each step of generation, we propose an *adaptive rejection sampling algorithm* that typically requires orders of magnitude fewer constraint evaluations. Second, we show how this algorithm can be extended to produce low-variance, unbiased estimates of importance weights at a very small additional cost—estimates that can be soundly used within previously proposed *sequential Monte Carlo algorithms* to correct for the myopic behavior of local constraint enforcement. Through extensive empirical evaluation in text-to-SQL, molecular synthesis, goal inference, pattern matching, and JSON domains, we show that our approach is superior to state-of-the-art baselines, supporting a broader class of constraints and improving both runtime and performance. Additional theoretical and empirical analyses show that our method's runtime efficiency is driven by its dynamic use of computation, scaling with the divergence between the unconstrained and constrained LM, and as a consequence, runtime improvements are greater for better models.

 https://github.com/genlm/genlm-control

## 1 Introduction

Many tasks in scientific and engineering disciplines can be approached through controlled generation of strings from a language model subject to hard constraints. For example, we may seek to generate an API call that matches an endpoint, produce SQL queries that are consistent with the schema of a database, or design a molecule that satisfies a target specification. The dominant approach in these settings is **locally constrained decoding** (**LCD**), which forces each sampled token to conform to the constraint (Lu et al., 2021; Shin et al., 2021; Scholak et al., 2021; Lu et al., 2022; Poesia et al., 2022; Shin & Van Durme, 2022; Geng et al., 2023; Beurer-Kellner et al., 2024; Huang et al., 2024; Moskal et al., 2024; Ugare et al., 2024; Wang et al., 2024a; Zheng et al., 2024; Banerjee et al., 2025).[1]

---

[*]co-first authorship, [‡]co-senior authorship. contact: `lipkinb@mit.edu` & `tim.f.vieira@gmail.com`

[1]This paper focuses on *runtime* control, but we note that many *training-based* control methods exist (e.g., Ziegler et al., 2019; Stiennon et al., 2020; Bai et al., 2022; Ouyang et al., 2022; Rafailov et al., 2023).

This approach suffers from two critical drawbacks. First, the typical method for sampling from this distribution by enumerative **token masking** can be *slow*. Full masking requires checking the constraint against every item in the vocabulary—which can often consist of more than $100,000$ tokens—before finally filtering, renormalizing, and sampling. In some special cases, such as when the constraint can be expressed as a regular or context-free grammar, optimizations are available that make checking every token feasible (Willard & Louf, 2023; Kuchnik et al., 2023; Koo et al., 2024; Ugare et al., 2024; Dong et al., 2024; Cognetta et al., 2025; Park et al., 2025). However, for **black-box constraints**, such optimizations do not apply. Second, as often observed (e.g., Lew et al., 2023; Park et al., 2024; Gareev et al., 2024; Ahmed et al., 2025; Loula et al., 2025; Kempton & Burrell, 2025; Ye et al., 2025), by renormalizing only the *local* distribution (over tokens), this approach can distort the *global* distribution (over strings). Because LCD myopically enforces constraints at each step, it can sometimes greedily sample its way into low-probability regions of sequence space (see §2 for discussion).

Several papers have noted this problem and proposed approaches that recast controlled generation as probabilistic conditioning, treating the problem as posterior inference (Rosenfeld et al., 2001; Miao et al., 2020; Krause et al., 2021; Yang & Klein, 2021; Meng et al., 2022; Qin et al., 2022; Shih et al., 2023; Zhang et al., 2023; Hu et al., 2024; Zhang et al., 2024a; Faria & Smith, 2025). In this framework, approximate inference methods—e.g., importance sampling (IS) or sequential Monte Carlo (SMC)—can be used to correct sampled sequences from the locally constrained distribution to the target global posterior (Börschinger & Johnson, 2011; Dubbin & Blunsom, 2012; Yang & Eisenstein, 2013; Buys & Blunsom, 2015; Lin & Eisner, 2018; Lew et al., 2023; Zhao et al., 2024; Puri et al., 2025; Loula et al., 2025).[2]

A key diagnostic quantity for assessing the quality of a generated sequence is the marginal probability of the constraint under the LM's local next-token distribution at each timestep, $Z \stackrel{\text{def}}{=} \sum_{x \in \mathcal{X}} p_0(x) \mathbb{1}_\mathcal{C}(x)$, where $p_0$ is the LM next-token-distribution over the vocabulary $\mathcal{X}$, and $\mathbb{1}_\mathcal{C}$ checks whether a given token conforms to the constraint at the current time step. Within sequential IS or SMC, $Z$ arises as an incremental importance weight update. When this quantity is low, we have reached a point in generation where it is difficult to sample any continuation of the sequence that satisfies the constraint. For SMC specifically, this can be used as a signal for reallocating computation to more promising sequences during subsequent resampling. Using token masking, it is clear how to compute $Z$—simply sum over all unmasked tokens. However, if we would like to sample from the local constrained token distribution without evaluating $\mathbb{1}_\mathcal{C}$ over the entire vocabulary, this quantity may not be directly available, and we must seek to estimate it.

In this paper, we introduce an exact sampler for constrained token distributions whose runtime is faster than token masking, often by orders of magnitude. Our algorithm is based on **rejection sampling** and, thus, is a **Las Vegas algorithm**, i.e., an exact algorithm with a stochastic runtime. Rather than using fixed compute for each step, compute scales dynamically based on need. Furthermore, we derive unbiased estimators of $Z$ for this algorithm that can be used to correct samples globally within an SMC framework.

Our core contributions are as follows:

- **A fast Las Vegas sampling algorithm compatible with any constraint.** We develop *adaptive weighted rejection sampling* (AWRS), a sampling algorithm for constrained generation. Like other rejection sampling algorithms, the cost of our approach scales with the difficulty of the constraint. However, our adaptive algorithm outperforms standard rejection sampling by capitalizing on the statistical structure of the next token distribution. Due to the speed of this algorithm, we can comfortably evaluate arbitrary black-box constraints.
- **Stochastic estimates of $Z$ to correct for greediness.** We develop an approach to calculate low-variance, unbiased estimates of $Z$ for AWRS, supporting its integration within approximate inference methods like SMC.
- **Runtime analysis.** We theoretically and empirically characterize the work done by constrained decoding in terms of the probabilistic update between the LM's unconditional next token distribution and the conditioned target distribution. Critically, AWRS runs

---

[2]See App. A for a detailed discussion of language modeling as probabilistic conditioning, importance sampling, and sequential Monte Carlo.

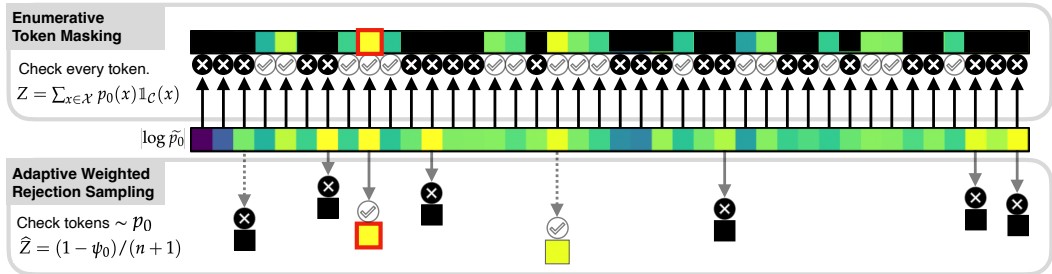

Figure 1: Our approach (bottom) compared to enumerative token masking (top). Only a subset of the tokens are checked while sampling from the same distribution.

faster when the base model is better able to capture the constraint—meaning that our approach is more efficient for more accurate LMs—aligning with scaling trends.

- **Empirical evaluations.** We evaluate AWRS alongside several state-of-the-art baselines on five challenging controlled generation benchmarks. AWRS yields improvements to expressiveness, runtime, and accuracy across relevant comparisons.

## 2 Background

**Locally Constrained Decoding (LCD).** A **language model** $P_0$ over an **alphabet** $\mathcal{X}$ is a probability distribution over the set of **strings** $\mathcal{X}^*$. We write $x$ for individual tokens and $\boldsymbol{x}$ for complete strings. A **global constraint function** $\mathbb{1}_{\mathcal{L}} \colon \mathcal{X}^* \to \{0,1\}$ encodes the **set of valid strings** $\mathcal{L} \subseteq \mathcal{X}^*$. The problem of controlled generation can be expressed as sampling $\boldsymbol{x} \sim P$, where $P \overset{\text{def}}{=} P_0(\cdot \mid \boldsymbol{x} \in \mathcal{L})$, i.e., sampling a complete string from the LM that satisfies the constraint while preserving the relative probabilities of possible strings. This latter property is what distinguishes proper global conditioning from arbitrary constrained sampling.

During the autoregressive generation of a string, it is sometimes possible to evaluate whether sampling any given token would cause an immediate violation of the constraint. For example, if a constraint requires writing a sentence where no word exceeds 5 characters, the only possible continuations to "north" should induce word boundaries, rather than continuing with "ern" or "east". At each step of generation, based on a current string prefix $\boldsymbol{x}_p$, we assume access to a **local constraint function** $\mathbb{1}_{\mathcal{C}} \colon \mathcal{X} \to \{0,1\}$, encoding a **set of valid next tokens** $\mathcal{C} \subseteq \mathcal{X}$, such that $x \notin \mathcal{C} \implies \forall \boldsymbol{x}_s \colon \boldsymbol{x}_p \, x \, \boldsymbol{x}_s \notin \mathcal{L}$. That is, if the local constraint function rejects a token $x$, there is no valid continuation of $\boldsymbol{x}_p$ that begins with $x$.[3]

Fixing a prefix $\boldsymbol{x}_p$, let $p_0$ denote the LM's **local prior distribution** over the next token—note this distribution is conditioned on the preceding tokens. At each step, LCD samples a next token from the **local posterior distribution** $p$ induced by the local constraint function:

$$p(x) \overset{\text{def}}{=} \frac{1}{Z}\widetilde{p}(x) \qquad \widetilde{p}(x) \overset{\text{def}}{=} p_0(x)\,\mathbb{1}_{\mathcal{C}}(x) \qquad Z \overset{\text{def}}{=} \sum_{x \in \mathcal{X}} \widetilde{p}(x) \tag{1}$$

where $\widetilde{p}$ is its unnormalized density and $Z$ is its normalizing constant. (Note $0 < Z \le 1$.) Intuitively, $Z$ is the total probability of satisfying the local constraint under $p_0$. The most common approach to implementing LCD is token masking: computing $p$ exactly during generation, by evaluating $\mathbb{1}_{\mathcal{C}}$ on every item in the vocabulary, zeroing impossible tokens, summing $p_0(x)$ over the remainder to compute $Z$, and renormalizing to obtain $p$ (Fig. 1, top).

By applying constraints locally at every token, LCD can greedily sample its way into low-probability sequences that are difficult or impossible to recover from. For example, continuing with the constraint of generating a sentence where no word may exceed 5 characters, an LM will struggle to complete the prefix "The Fed says the cost of a 30-yr fixed mortg", despite there having been no issues up until that point (see Lew et al., 2023).

---

[3]See App. I.2 for a more detailed discussion of mapping global constraints to local constraint checkers.

Interestingly, such dead-ends can be readily detected using precisely the quantity $Z$ computed as a byproduct of LCD. When $Z$ is very low, we have reached a point in string generation where only very few, improbable tokens can continue the string.[4] This local $Z$ can be used in **sequential Monte Carlo (SMC)** to reweight partial generations (particles) for subsequent re-sampling steps, which tend to eliminate unpromising string prefixes and duplicate more promising ones (App. A and Alg. 1). In this setting—where LCD is used as a **proposal distribution** within SMC—$Z$ is precisely the incremental correction factor that can be derived by simplifying the importance weights (Naesseth et al., 2019).

Since $Z$ is computed as a byproduct of token masking, this algorithm provides its own correction factors when used as a part of a proposal distribution in SMC. Unfortunately, however, modern LMs often have vocabularies exceeding $100,000$ tokens; evaluating $\mathbb{1}_\mathcal{C}$ on every token in this set—and, thus, exactly computing $Z$—is often prohibitively slow. It would initially appear that exact sampling from the local posterior $p$ and exact weight correction with $Z$ are impractical in these cases.

In the next sections, we develop an approach to sampling from the local posterior $p$ that is exact, tractable for large vocabularies, typically fast in terms of average time, and provides correction factors to correct the greediness of LCD.

**Simple rejection sampling.**  Simple rejection sampling is an algorithm that can provide exact samples from $p$ without looping through the entire token vocabulary. It works by drawing a token $x \sim p_0$, then checking whether the token satisfies $\mathbb{1}_\mathcal{C}(x)$. If it does, the token is returned; otherwise, this process is repeated. The returned token is an exact sample from $p$.

Compared to the constant cost of token masking, this algorithm is a **Las Vegas algorithm**, i.e., an exact algorithm with a stochastic runtime. It is easy to show that the expected number of samples drawn by the algorithm before meeting the constraint is $\frac{1}{Z}$; it can furthermore be shown that $D_{KL}\left(p \parallel p_0\right) = -\log Z$ (Levy, 2008; Freer et al., 2010), and thus that the expected runtime is exponential in $D_{KL}\left(p \parallel p_0\right)$. When the constraint is relatively easy to satisfy (i.e., when $Z \approx 1$ so $D_{KL}\left(p \parallel p_0\right)$ is low), this can lead to runtimes that are much faster than those required by full-vocabulary token masking. However, when $Z$ is very small ($Z < |\mathcal{X}|^{-1}$), rejection sampling's runtime can be even worse than that of token masking.

**Adaptive rejection sampling (ARS).**  Adaptive rejection sampling (Gilks & Wild, 1992; Mansinghka et al., 2009) is a version of rejection sampling that is never slower than token masking and often significantly faster. In ARS, we adaptively *remove* each invalid token encountered while rejection sampling, so that we never sample the same rejected token twice. This simple modification can dramatically improve the runtime of the algorithm (Fig. F.2).

Both rejection sampling approaches evaluate $\mathbb{1}_\mathcal{C}$ only on the samples they draw and, thus, can be more efficient than token masking. However, neither provides a direct way to exactly compute $Z$. A solution to this problem arises from the observation that for SMC, it suffices to use independent unbiased estimates of each local $Z$, rather than the exact quantities, as correction factors (App. A and Alg. 2; Naesseth et al., 2015).

## 3   Our Algorithms

In lieu of the exact but expensive correction factors obtained as a byproduct of token masking, we wish to find cheap, unbiased estimates of $Z$ that we can compute as a byproduct of simple or adaptive rejection sampling.

Our first observation toward this goal is that the *number of rejected tokens* sampled before generating a valid next token $x$ contains signal about the magnitude of $Z$: when many

---

[4]Why $Z$ and not sequence probability? While methods optimizing sequence probability, like beam search, partially address the sample quality desiderata, there are caveats. If there are many valid tokens, whose sum is large, but each individual token has low probability, sequence probability will unfairly penalize any token (Koehn & Knowles, 2017; Murray & Chiang, 2018; Cohen & Beck, 2019; Stahlberg & Byrne, 2019). Methods that account for $Z$ do not suffer this drawback.

rejected tokens are generated, this suggests $Z$ is small, and can serve as a sign that the current prefix $x_p$ has landed us in a low-probability region of the global posterior $P$.

Indeed, in simple rejection sampling, the number of trials $n_0 + 1$ (that is, $n_0$ rejections plus the final success) is an unbiased estimate of $1/Z$. Unfortunately, $1/(n_0 + 1)$ is *not* an unbiased estimate of $Z$, and we require unbiased estimates of $Z$ for sound use within SMC.

## 3.1 Warm-up: Weighted Rejection Sampling (WRS)

In the simple rejection sampling setting, we can collect more data about $Z$ by running $L \geq 1$ *additional* rejection loops. In addition to reducing variance, this also supports calculation of an *unbiased* estimate of $Z$. The total number of trials $T$ (rejections and successes) required to reach $L + 1$ successes follows a negative binomial distribution with parameters $Z$ and $L + 1$. Letting $n = \sum_{i=0}^{L} n_i$ be the total number of rejections across the $L + 1$ loops,

$$\widehat{Z} \stackrel{\text{def}}{=} \frac{L}{T-1} = \frac{L}{(n + (L+1)) - 1} = \frac{L}{n + L} \tag{2}$$

is the minimum variance unbiased estimator (MVUE) for the $Z$ parameter of the negative binomial distribution, provided that $L \geq 1$ (Forbes et al., 2011, App. C.1). This allows us to define the following algorithm for jointly generating a next token $x$ from $p$ alongside an unbiased estimate $\widehat{Z}$ of $Z$.

**Definition 1.** *Given an unnormalized target $\widetilde{p}$ as above, **weighted rejection sampling** generates $\langle x, \widehat{Z} \rangle \sim Q_{WRS}$ as follows:*

1. *Run rejection sampling to obtain a valid sample $x$: Sample $r_1, \dots, r_{n_0}, x \stackrel{i.i.d.}{\sim} p_0$ through $n_0$ rejections $r_i$ until obtaining an accepted token $x \in \mathcal{C}$.*
2. *For a budget of $L \geq 1$ additional loops, repeat step 1 and count the number of rejections on each loop $n_1, \dots, n_L$.*
3. *Calculate the estimate $\widehat{Z} \stackrel{\text{def}}{=} \frac{L}{n+L}$, where $n = \sum_{i=0}^{L} n_i$.*
4. *Return $\langle x, \widehat{Z} \rangle$*

*An implementation in NumPy can be found in Listing 1.*

**Proposition 1.** *For $\langle x, \widehat{Z} \rangle \sim Q_{WRS}$, $x$ is distributed according to $p$ and $\mathbb{E}[\widehat{Z}] = Z$.*     *(App. C)*

**Proposition 2.** *The expected runtime of $Q_{WRS}$ scales with $\mathcal{O}(\frac{L}{Z})$.*     *(App. G.1)*

Using these estimates, simple rejection sampling may be soundly integrated into SMC as a proposal distribution, supporting the correction of LCD's greediness. $L = 1$ is enough to ensure unbiasedness, and we find it to work well in practice, but $L$ can be increased to trade higher runtime for reduced variance (Fig. F.1).

## 3.2 Adaptive Weighted Rejection Sampling (AWRS)

In the adaptive setting, the expected number of rejections is reduced, and the negative binomial estimator can no longer be used. However, an auxiliary-variable argument based on the framework of Lew et al. (2022) can be used to derive an alternative formula for the adaptive case based on not just the number of rejections but also the probability mass removed from $p_0$ during adaptation.

**Definition 2.** *Given an unnormalized target $\widetilde{p}$, AWRS generates $\langle x, \widehat{Z} \rangle \sim Q_{AWRS}$ as follows:*

1. *Sample $\langle r_1, \dots, r_{n_0}, x \rangle$ as follows: draw $n_0$ unique rejections $r_i$ until obtaining $x \in \mathcal{C}$. Note that beyond the first step, we do not sample from $p_0$, but a re-normalized distribution on $\mathcal{X} \setminus \mathbf{r}_{<i}$.*
2. *Calculate $\psi_0 = \sum_{i=1}^{n_0} p_0(r_i)$.*
3. *Generate one additional trace $\langle s_1, \dots, s_{n_1}, x^* \rangle$, by continuing to sample as above from the remaining not-yet-rejected elements, through an additional $n_1$ new unique rejections, until finding an element $x^*$. Note that $x^*$ could be the same as $x$, since acceptances are replaced (unlike rejections).*
4. *Calculate the estimate $\widehat{Z} \stackrel{\text{def}}{=} \frac{1 - \psi_0}{n+1}$, where $n = n_0 + n_1$.*

*5. Return $\langle x, \widehat{Z} \rangle$*

*An implementation in NumPy can be found in Listing 2.*

**Proposition 3.** *For $\langle x, \widehat{Z} \rangle \sim Q_{AWRS}$, $x$ is distributed according to p and $\mathbb{E}[\widehat{Z}] = Z$.*  *(App. D)*

**Proposition 4.** *The expected runtime of $Q_{AWRS}$ scales with $\mathcal{O}(\sum_{x \notin \mathcal{C}} \pi_x)$, where $\pi_x \overset{\text{def}}{=} \frac{p_0(x)}{p_0(x)+Z}$, i.e., the probability of each non-conforming token relative to Z.*  *(App. G.2)*

As above, AWRS generates next tokens and correction factors suitable for use within SMC. In addition, AWRS offers considerable runtime benefits. Trivially, since we cannot resample rejections, we must succeed after at most $|\mathcal{X} \setminus \mathcal{C}|$ rejection steps—the number of invalid tokens—no matter how small Z is. AWRS also has lower *expected* runtime. Intuitively, we may think of the time it takes to sample an acceptance as follows. Each non-conforming token may be considered a *distractor* if its *individual* mass is comparable to or higher than Z, the sum of *all* conforming tokens. Rather than all non-conforming tokens contributing equally, expected runtime is dominated by only these—typically rare—distractor tokens. The exact value is derived and further explored in App. G.2. We also propose several extensions of AWRS with beneficial properties in App. H, including support for partial concurrency, properly weighted approaches to early stopping based on either mass explored or steps taken, and methods for combination with truncation sampling.

## 4 Experiments

Our experiments measure the practical impact of our methods on accuracy and runtime for 5 tasks in different domains.[5] We use task-specific metrics rather than evaluating the sampler's internal behavior (e.g., how accurately it estimates Z), but see Fig. 2 and App. F. We compare AWRS to strong baselines through consideration of two versions: ARS-LCD, which performs simple LCD using *unweighted* adaptive rejection sampling, and AWRS-SMC, which uses the weighted version within SMC.

**Methods.**  We first compare the following *uncorrected* methods in terms of both runtime and downstream task accuracy. Methods in this section yield $M = 1$ unweighted samples:

- **Base language model (Base LM).** Sample sequences of tokens from the language model (without forcing any constraints), i.e., sample $x \sim P_0$.
- **Locally constrained decoding with token masking (TM-LCD).** The standard approach to constrained decoding. We mask the entire token vocabulary, re-normalize, and sample.[6]
- **Locally constrained decoding with adaptive rejection sampling (ARS-LCD).** A faster implementation of LCD: rather than masking the entire vocabulary, we draw a sample from the same LCD distribution using ARS. (We do not yet use an importance-weight correction, so we run only the first rejection loop.)

The baselines above allow us to gauge the degree to which adaptive rejection sampling improves runtime. Our next set of methods go beyond LCD, returning a weighted ensemble of $M$ strings such that the expected weight of $x$ in the ensemble is $P_0(x) \cdot \mathbb{1}_{\mathcal{L}}(x) \propto P(x)$.

- **Sample-Verify.** Sample $M$ complete strings from the LM and weight them according to $\mathbb{1}_{\mathcal{L}}$ (which amounts to discarding strings $x \notin \mathcal{L}$).[7]

---

[5]We note that our implementations are all written in pure Python and are relatively unoptimized. Runtime improvements presented in this work are driven purely by algorithmic advancement. Being a flexible plug-and-play method, we encourage talented systems practitioners to capture the many untapped speedups.

[6]Due to the high cost of full token masking, we only include this baseline for one benchmark, from which we illustrate our orders-of-magnitude speed-up.

[7]This baseline is a common approach for incorporating constraints into an LM generation pipeline (Cobbe et al., 2021; Hendrycks et al., 2021; Nakano et al., 2021; Ahn et al., 2022; Shi et al., 2022; Uesato et al., 2022; Olausson et al., 2023; Lightman et al., 2023; Ankner et al., 2024; Gandhi et al., 2024; Wang et al., 2024b; Xin et al., 2024; Zhang et al., 2024b).

- **Sequential Monte Carlo with constraint as twist (Twisted SMC).** Sample tokens directly from the LM, but use $\mathbb{1}_{\mathcal{C}}(x)$ as a **twist function** to filter partial sequences after they have been extended with a token $x$. Note that this is a programmable twist as in Loula et al. (2025), rather than a learned twist (Naesseth et al., 2019; Lawson et al., 2022).
- **Sequential Monte Carlo with AWRS proposal (AWRS-SMC).** Use the AWRS algorithm as a proposal distribution for SMC. Like ARS-LCD, this method generates tokens using an adaptive rejection sampling loop, but *does* calculate the correction factor.

**Metrics.**

- **Accuracy.** The accuracy of a returned string is defined by the benchmark. For methods that construct a weighted ensemble, we report the expected accuracy of a random string returned from this ensemble (with probability proportional to its weight).[8]
- **Runtime.** The average number of seconds it takes to generate the $M$ complete strings.[9]

**Benchmarks.**

- **Text-to-SQL (Spider).** *Task:* Generate SQL queries from a natural language question paired with its corresponding database schema. *Data:* Development split of the Spider dataset (Yu et al., 2018). *Metric:* Execution accuracy (checking if the produced SQL query, when executed on a test database, yields the same results as the ground-truth query). *Base LM:* Llama 3.1 8B-Instruct. *Constraint function:* A Python parser for the SQL context-free grammars provided by Roy et al. (2024) to enforce syntactically valid SQL.
- **JSON.** *Task:* Generate documents that conform to a specific JSON Schema. *Data:* The validation splits of the Github-trivial, -easy and -medium tasks in the JSONSchemaBench dataset (Geng et al., 2025). *Metric:* Whether a valid JSON document conforming to the schema is generated. *Base LM:* Llama 3.1 8B-Instruct. *Constraint function:* Check the output parses as JSON, and validate using the Python jsonschema library. Parsing is done with a streaming JSON parser, which allows incremental detection of some schema violations before the full document has been generated.
- **Goal inference (Planetarium).** *Task:* Formally define an agent's goal within the STRIPS subset of the PDDL planning language, using a natural-language description of the goal alongside PDDL code that specifies the agent's starting conditions and plan to reach it. *Data:* Blocksworld tasks featuring up to 10 objects from the Planetarium benchmark (Zuo et al., 2024). *Metric:* Equivalence to the ground-truth PDDL description. *Base LM:* Llama 3.1 8B. *Constraint function:* Check STRIPS syntax for goals as defined in the Planetarium Blocksworld domain + execute a simulation using a ground-truth plan to verify if the resulting state matches the predicted (partial) goal.
- **Molecular Synthesis** *Task:* Produce drug-like compounds using the SMILES notation (Weininger, 1988). *Data:* Few-shot prompts created by repeatedly selecting 20 random samples from the GDB-17 database (Ruddigkeit et al., 2012). *Metric:* Quantitative Estimate of Drug-likeness (QED; Bickerton et al., 2012), a widely used measure of molecular quality. *Base LM:* Llama 3.1 8B. *Constraint function:* A SMILES prefix validator implemented via the Python *partialsmiles* library (O'Boyle, 2024).
- **Pattern matching.** *Task:* Generate strings that conform to expressive pattern-matching specifications. Compared to formal regular expressions, these patterns contain explicit features that cannot be fully captured by deterministic finite-state automata, including unbounded center embedding and conditionals. *Data:* Over 400 pattern-matching specifications generated via the pipeline in App. J. *Base LM:* Llama 3.1 8B-Instruct. *Metric:* Adherence to the specified pattern. *Constraint function:* An incremental pattern validator that checks whether a complete match remains possible given a prefix (Barnett, 2014).

---

[8] The rationale is that the probability of returning $x$ then approaches $P(x)$ as $M$ grows, so we approximately return $x \sim P$, just as Base LM returns $x \sim P_0$. Note that we could plausibly improve accuracy further by selecting the most probable string from the ensemble, or more generally, by the **minimum Bayes risk** method of selecting or constructing a "consensus string" with low expected task loss under the weighted ensemble.

[9] Our runtimes scale sublinearly in $M$ because we use parallel hardware (a GPU). Specifically, the calls to obtain the next-token distribution $p_0$ from the LLM are batched over the $M$ strings.

| Method | Accuracy | Runtime (sec/ex) |
|---|---|---|
| Base LM | 0.523 (0.50, 0.54) | 0.78 (0.76, 0.80) |
| ARS-LCD | 0.572 (0.55, 0.59) | 1.02 (0.98, 1.06) |
| Sample-Verify | 0.609 (0.59, 0.63) | 2.71 (2.61, 2.82) |
| Twisted SMC | 0.608 (0.59, 0.63) | 2.60 (2.48, 2.72) |
| AWRS-SMC | 0.600 (0.58, 0.62) | 3.02 (2.90, 3.14) |

(a) Text-to-SQL

| Method | Accuracy | Runtime (sec/ex) |
|---|---|---|
| Base LM | 0.688 (0.66, 0.72) | 2.35 (2.20, 2.52) |
| ARS-LCD | 0.773 (0.74, 0.80) | 4.41 (4.08, 4.76) |
| Sample-Verify | 0.858 (0.83, 0.88) | 6.18 (5.81, 6.56) |
| Twisted SMC | 0.871 (0.84, 0.90) | 5.17 (4.65, 5.71) |
| AWRS-SMC | 0.898 (0.87, 0.92) | 12.71 (11.58, 13.87) |

(b) JSON

| Method | Accuracy | Runtime (sec/ex) |
|---|---|---|
| Base LM | 0.032 (0.01, 0.06) | 1.07 (0.97, 1.17) |
| ARS-LCD | 0.18 (0.11, 0.26) | 0.77 (0.68, 0.86) |
| Sample-Verify | 0.205 (0.13, 0.28) | 4.55 (4.25, 4.84) |
| Twisted SMC | 0.479 (0.39, 0.57) | 3.20 (2.93, 3.47) |
| AWRS-SMC | 0.528 (0.44, 0.62) | 2.62 (2.42, 2.82) |

(c) Goal Inference

| Method | Accuracy | Runtime (sec/ex) |
|---|---|---|
| Base LM | 0.271 (0.25, 0.29) | 0.69 (0.67, 0.71) |
| ARS-LCD | 0.522 (0.51, 0.54) | 0.97 (0.93, 1.02) |
| Sample-Verify | 0.594 (0.59, 0.60) | 1.83 (1.79, 1.86) |
| Twisted SMC | 0.591 (0.59, 0.60) | 1.53 (1.51, 1.56) |
| AWRS-SMC | 0.615 (0.61, 0.62) | 3.41 (3.31, 3.50) |

(d) Molecular Synthesis

| Method | Accuracy | Runtime (sec/ex) | Method | Accuracy | Runtime (sec/ex) |
|---|---|---|---|---|---|
| Base LM | 0.562 (0.51, 0.61) | 0.10 (0.09, 0.11) | Sample-Verify | 0.786 (0.75, 0.83) | 0.26 (0.23, 0.29) |
| ARS-LCD | 0.980 (0.97, 0.99) | 0.16 (0.13, 0.20) | Twisted SMC | 0.813 (0.77, 0.85) | 0.19 (0.18, 0.21) |
| TM-LCD | 0.978 (0.96, 0.99) | 6.91 (5.68, 8.46) | AWRS-SMC | 0.990 (0.98, 1.00) | 0.43 (0.35, 0.52) |

(e) Pattern Matching

Table 1: Comparison of method accuracy and runtime across domains with 95% bootstrapped confidence intervals. Runtime represents the average execution time (in seconds) across all instances in the dataset. Sample-Verify and Twisted SMC were run with $M = 10$ particles. AWRS-SMC was run with $M = 5$ particles.

The runtime costs associated with each benchmark's constraint checker, along with further practical discussion of constraint programming, can be found in App. I.1.

## 5 Results & Discussion

**AWRS Outperforms State-of-the-Art Controlled Generation Methods.** Tab. 1 shows the accuracy and runtime of each method in each domain. We observe the following results:

- **Controlled generation outperforms uncontrolled generation.** With little overhead to runtime, ARS-LCD improves accuracy over Base LM across all benchmarks.
- **Adaptive sampling is much faster than token masking, with no loss of accuracy.** On the pattern matching domain — the only one where it was computationally feasible to run TM-LCD — ARS-LCD matches its accuracy while being $> 50\times$ faster.[10]
- **Correcting for greediness improves accuracy.** AWRS-SMC always matches or beats ARS-LCD, significantly improving it in four domains (Goal Inference, JSON, Text-to-SQL, Molecular Synthesis). The other domain (Pattern Matching) suffers somewhat less under greediness because its local constraint $\mathbb{1}_\mathcal{C}$ is more precise, allowing a prefix only if it has a valid continuation.
- **AWRS-SMC outperforms existing approaches that correct for greediness.** With half the number of particles, AWRS-SMC attains accuracy comparable to or higher than Sample-Verify and Twisted SMC in all benchmarks.

Next, we investigate how AWRS-SMC scales with LM size (Fig. L.1) compared to existing methods that sample from the global distribution.

- **AWRS-SMC with smaller LMs outperforms existing SMC approaches with larger LMs.** Fig. L.1 shows how AWRS-SMC using Llama 3.2 1B yields better runtime and accuracy than Twisted SMC using Llama 3.1 8B and Llama 3.3 70B. This suggests that including

---

[10]This $50\times$ speedup is at the level of complete sequence generation, including all time spent on LM computation. The speedup factor modulo this constant is much greater.

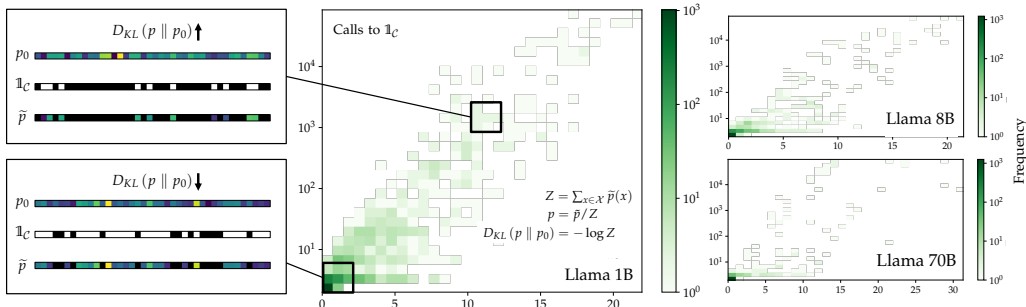

Figure 2: The number of AWRS calls to $\mathbb{1}_{\mathcal{C}}$ (**y-axis**) scales with $D_{KL}\left(p \parallel p_0\right)$ (Nats; **x-axis**).

informative constraints in the proposal is a compute-efficient way to make small models punch above their weight.

**AWRS Achieves Speedups by Allocating Computation Dynamically.** As shown through our extensive theoretical and empirical runtime analyses (App. G.2 and Figs. G.1 and F.2), AWRS scales with the difficulty of constraint conformance $D_{KL}\left(p \parallel p_0\right)$, taking less time to sample a token when the constraint is expected and more time when it is not. We analyzed the TM-LCD results of the pattern-matching benchmark, where token masking supports the exact calculation of the ground truth $D_{KL}\left(p \parallel p_0\right)$ for each token sampling step. We then ran AWRS for each of these steps, illustrating a few key results (Fig. 2).

1. $D_{KL}\left(p \parallel p_0\right)$ is small in most sampling steps; AWRS typically checks only 2 or 3 tokens.
2. As $D_{KL}\left(p \parallel p_0\right)$ increases for the hardest cases, the runtime of AWRS scales dynamically. An interesting consequence is that *AWRS samples faster for more accurate base models*.
3. As $D_{KL}\left(p \parallel p_0\right)$ grows, AWRS generally does not deteriorate. AWRS is roughly bounded by the number of non-conforming tokens whose individual probabilities are close to or exceed $Z$. This set is typically small, and it turns out empirically that more accurate models seem to reduce the size of this set. Even when a model's top choice is wrong, it often still prefers constraint-conforming tokens to arbitrary non-conforming tokens.

# 6 Related Work

One approach to accelerating constrained decoding has been to pre-compile restricted constraint classes to reduce runtime overhead. Engineering advances have enabled at least partial compilation of constraints expressible as membership in regular (Deutsch et al., 2019; Willard & Louf, 2023; Kuchnik et al., 2023) or context-free (Koo et al., 2024; Ugare et al., 2024; Dong et al., 2024) languages as well as restricted classes of Boolean circuits (Ahmed et al., 2025). In comparison, our approach supports arbitrary programmable constraints.

Another approach has explored limiting the number of constraint evaluations. Poesia et al. (2022) and Ugare et al. (2025) allow the LM to proceed unrestrictedly and then backtrack on errors. Scholak et al. (2021) and Shin & Van Durme (2022) use top-*k* truncation within beam search. Loula et al. (2025), in a similar spirit to Morin & Bengio (2005), hierarchically stratify the vocabulary and incrementalize the constraint checker to byte sequences. A subset of the Outlines library (v.0.2.2; Willard & Louf, 2023) uses a variant of deterministic probability-ordered search, first sorting tokens by their logits and then yielding the first conforming token. Botta et al. (2025) and Mündler et al. (2025) propose related variants of simple unweighted token-level rejection sampling. Our work builds on these approaches within a probabilistic framework, deriving an exact sampler equivalent to token masking (ARS-LCD) alongside importance weights supporting the sound use of such samples in SMC (AWRS-SMC).

Several recent papers have used SMC to correct the greediness of LCD (Lew et al., 2023; Zhao et al., 2024; Loula et al., 2025). These approaches come with significant limitations: Zhao et al. (2024) require an expensive fine-tuning procedure for learning twists, whereas Loula et al. (2025) require constraints to be decomposable into slow and fast components,

the latter of which must still be evaluated over large sets of tokens. In contrast, our approach works with any constraint out of the box and evaluates it frugally, being typically more accurate than twisted SMC for any fixed runtime.

## 7 Conclusion

Locally constrained decoding is both slow and greedy. In this paper, we address these weaknesses respectively by introducing (i) an adaptive rejection sampler requiring orders of magnitude fewer constraint evaluations, and (ii) an algorithm computing weights for transforming this local sampler into a global one. Across many challenging controlled generation domains, we find that our method is faster and more accurate than existing methods, even when using fewer particles and smaller LMs. Finally, we use theoretical and empirical analyses to show that the number of constraint evaluations our method makes scales with the KL divergence between the unconstrained and constrained LM distributions—as a consequence, our method is faster for more powerful LMs.

## Reproducibility

The most up-to-date and performant AWRS-SMC reference implementation can be found in the following actively maintained library: https://github.com/genlm/genlm-control

The source code and data to replicate this paper's experiments can be found in the following repository: https://github.com/genlm/awrs-colm-2025

## Author Contributions

**First Authors**
- **Benjamin Lipkin** (lipkinb@mit.edu): research conception, formal analysis (algorithms, proper weighting), software development (samplers), experiment development (simulation, pattern matching), visualization, writing
- **Benjamin LeBrun** (benjamin.lebrun@mail.mcgill.ca): lead software engineer (infrastructure, interfaces, SMC), experiment development (text-to-SQL, JSON, pattern-matching), visualization, writing

**Contributors**
- **Jacob Hoover Vigly** (jahoo@mit.edu): formal analysis (runtime complexity, parameter estimator), technical advice (sampling algorithms), visualization, writing
- **João Loula** (jloula@mit.edu): software development (algorithm optimization), experiment development (goal inference, molecular synthesis), visualization, writing
- **David R. MacIver** (david@drmaciver.com): experiment development (JSON) and bounded cost extensions to AWRS.
- **Li Du** (leodu@cs.jhu.edu): software development (grammar parsing prototype)
- **Jason Eisner** (jason@cs.jhu.edu): technical advice (sequential inference), writing
- **Ryan Cotterell** (ryan.cotterell@inf.ethz.ch): organization management, writing
- **Vikash Mansinghka** (vkm@mit.edu): organization management

**Senior Authors**
- **Timothy J. O'Donnell** (timothy.odonnell@mcgill.ca): organization management, senior project leadership, project narrative development, writing
- **Alexander K. Lew** (alexander.lew@yale.edu): senior project leadership, research conception, formal analysis (proper weighting), project narrative development, project advising and mentorship, writing
- **Tim Vieira** (tim.f.vieira@gmail.com): senior project leadership, research conception, formal analysis (algorithms, sequential inference), software development (grammar parsing), project narrative development, project advising and mentorship, writing

## Acknowledgments

BLi is supported by a National Science Foundation Graduate Research Fellowship under Grant No. 2141064. JHV is supported by a National Science Foundation SBE Postdoctoral Research Fellowship under Grant No. SMA-2404644. This research was enabled in part by compute resources provided by Mila (mila.quebec).

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

# A Conditional Language Modeling

**Strings.** An **alphabet** $\mathcal{X}$ is a finite set of symbols. Let $\mathcal{X}^*$ denote the set of all strings formed from the symbols in $\mathcal{X}$. Let $\varepsilon$ denote the empty string. Let $|x|$ denote the length of $x \in \mathcal{X}^*$. We also define $\square \notin \mathcal{X}$ as a special **end-of-string** marker.

**Constraints.** Let $\mathbb{1}_{\mathcal{L}} \colon \mathcal{X}^* \to \{0,1\}$ be a **constraint function**, encoding a **set of valid strings** $\mathcal{L} \subseteq \mathcal{X}^*$. Let $\vec{\mathcal{L}} \stackrel{\text{def}}{=} \{x \mid x\,x' \in \mathcal{L}\}$, i.e., the set of all valid prefixes of strings in $\mathcal{L}$. We define the **incremental constraint function**:

$$\vec{\mathbb{1}}_{\mathcal{L}}(x' \mid x) \begin{cases} \mathbb{1}_{x \in \mathcal{L}} & \textbf{if } x' = \square \\ \mathbb{1}_{x\,x' \in \vec{\mathcal{L}}} & \textbf{otherwise} \end{cases} \tag{3}$$

**Language models.** A **language model** $P_0$ is a probability distribution over $\mathcal{X}^*$. The **prefix probability** $\vec{P}_0(x)$ is the probability that a string drawn from $P_0$ has $x$ as a prefix:

$$\vec{P}_0(x) \stackrel{\text{def}}{=} \sum_{x' \in \mathcal{X}^*} P_0(x\,x') \tag{4}$$

The **conditional prefix probability** is the probability that a string drawn from $P_0$ has the prefix $x\,x'$ given that it already has the prefix $x$ for any strings $x, x' \in \mathcal{X}^*$:

$$\vec{P}_0(x' \mid x) \stackrel{\text{def}}{=} \frac{\vec{P}_0(x\,x')}{\vec{P}_0(x)} \quad \text{and} \quad \vec{P}_0(\square \mid x) \stackrel{\text{def}}{=} P_0(x)/\vec{P}_0(x) \tag{5}$$

Then, the probability of $x$ may be factorized as

$$P_0(x) = \vec{P}_0(\square \mid x) \prod_{t=1}^{|x|} \vec{P}_0(x_t \mid x_{<t}) \tag{6}$$

**Global conditioning.** We build on recent work defining constrained generation as probabilistic conditioning (e.g., Börschinger & Johnson, 2011; Dubbin & Blunsom, 2012; Yang & Eisenstein, 2013; Buys & Blunsom, 2015; Lin & Eisner, 2018; Miao et al., 2020; Krause et al., 2021; Yang & Klein, 2021; Meng et al., 2022; Qin et al., 2022; Zhang et al., 2023; Amini et al., 2023; Hu et al., 2024; Lew et al., 2023; Du et al., 2024; Zhao et al., 2024; Park et al., 2024; Ahmed et al., 2025; Loula et al., 2025). In particular, we define the **posterior distribution** $P(x)$ of a language model **prior** $x \sim P_0$ subject to the condition $\mathbb{1}_{\mathcal{L}}(x)$, as

$$P(x) \stackrel{\text{def}}{=} \frac{1}{G}\widetilde{P}(x) \qquad \widetilde{P}(x) \stackrel{\text{def}}{=} P_0(x)\mathbb{1}_{\mathcal{L}}(x) \qquad G \stackrel{\text{def}}{=} \sum_{x \in \mathcal{X}^*} \widetilde{P}(x) \tag{7}$$

For $P$ to be a well-defined probability distribution, we require $G = \Pr_{x \sim P_0}[\mathbb{1}_{\mathcal{L}}(x)] > 0$. In other words, there must be a way to generate a constraint-conforming string from $P_0$, and $G$ tells us the probability of this event. Note that

$$P(x) = \Pr_{X \sim P_0}[X = x \mid X \in \mathcal{L}] \tag{8}$$

Probabilistic conditioning is an enticing method for conditional generation as it is the only method that preserves the relative probabilities of all events that satisfy the condition.

Note that $P$ is a language model, so it has conditional prefix probabilities $\vec{P}$. Unfortunately, however, $G$ and the conditional prefix probabilities are generally intractable to compute exactly as they involve intractable sums over all $x \in \mathcal{X}^*$ (Rosenfeld et al., 2001). This makes ancestral sampling, i.e., left-to-right sampling from the conditional prefix probability distribution, infeasible. In what follows, we will develop methods for (approximate) sampling from $P$.

**(Sequence-level) Rejection sampling.** The most straightforward algorithm for sampling from $P$ is rejection sampling. Note that this is the intention of sample-verify methods:

```
1 def rejection_sampling():
2     while True:
3         x ∼ P₀  # sample complete string from the prior
4         if 𝟙_L(x):  # only return the string if the condition is satisfied
5             return x
```

Rejection sampling has an expected runtime of $\mathcal{O}(1/G)$ per sample. So it is only practical in this setting when $G \approx 1$.

**Locally constrained decoding.** Locally constrained decoding (typically performed through token masking) defines a language model $\ell$ that attempts to approximate $P$ by approximating $\vec{P}$ with $\vec{\ell}(x' \mid x) \approx \vec{P}(x' \mid x)$ where

$$\vec{\ell}(x' \mid x) \stackrel{\text{def}}{=} \frac{\vec{P_0}(x' \mid x)\,\vec{\mathbb{1}_L}(x' \mid x)}{L(x)} \quad \text{and} \quad L(x) \stackrel{\text{def}}{=} \sum_{x' \in \mathcal{X} \cup \{\square\}} \vec{P_0}(x' \mid x)\,\vec{\mathbb{1}_L}(x' \mid x) \tag{9}$$

Prior work (e.g., Lew et al., 2023; Park et al., 2024; Ahmed et al., 2025) has shown that this local approximation can be very different (i.e., biased) from $P$ in both theory and practice. Fortunately, it is possible to improve this approximation (i.e., overcome this bias) with additional computation using the techniques that we describe next.

**Explanation for why locally constrained decoding is biased.** Although $\ell$ is an effective method for generating strings $x \sim \ell$ that satisfy $\mathbb{1}_L(x)$, it has a tendency to over-represent certain strings. We can quantify this through the string's relative probability

$$\rho(x) \stackrel{\text{def}}{=} \frac{P(x)}{\ell(x)} = \frac{1}{G}\frac{P_0(x)\,\mathbb{1}_L(x)}{\ell(x)} = \frac{1}{G}\prod_{t=1}^{|x|+1} L(x_{<t}) \tag{10}$$

because

$$P(x) = \frac{P_0(x)\,\mathbb{1}_L(x)}{G} \quad \text{and} \quad \ell(x) = \frac{P_0(x)\,\mathbb{1}_L(x)}{\prod_{t=1}^{|x|+1} L(x_{<t})} \tag{11}$$

In other words, if we compare samples from the local distribution $\ell$ to their relative probability in the global distribution $P$, the string $x$ will appear at a rate of $\rho(x)$ more if $\rho(x) < 1$ or less if $\rho(x) > 1$ than it should.

In this work, $L(x) \leq 1$ for all $x$.

In the expression for $\rho(x)$, we see the factor that actually depends on the string $x$ independent of $G$, which is fortunate for us as we cannot efficiently compute $G$. Let $w(x) = G\rho(x) = \prod_{t=1}^{|x|+1} L(x_{<t})$.

Clearly, $\prod_{t=1}^{|x|+1} L(x_{<t})$ is an important diagnostic for sample quality.

Operationally, when sampling from $\ell$, the cumulative product of local $L$'s is a useful signal for detecting low-quality samples.

Interestingly, the average weight is equal to $G$, i.e., $\mathbb{E}_{x \sim \ell}\left[\prod_{t=1}^{|x|+1} L(x_{<t})\right] = G$.

Although the expression for the bias does not care about the ordering, it's hard not to think about the operational view of ancestral sampling from $\ell$; in this view, the sample is created from left to right by sampling from the conditional token distributions. We say that our specific sample is distorted by the product of its local normalization constants. This quantity measures how much the constraints affected the sample. If the constraints only rule out very low (or even zero) probability tokens, then the distortion will be small as $L$ will be $\approx 1$.

This operational view motivates the use of sequential Monte Carlo (SMC). In the case of SMC, we compare samples of complete and incomplete strings of length $\leq t$, and resample

those that appear to be on track, i.e., favoring those that have higher intermediate weight values. Thus, if a particle was favored initially by the proposal distribution, it may be evicted later in favor of a replica of a higher-weight particle.

Below is a simple example illustrating global conditioning, which can be used to illustrate the difference with respect to local conditioning.

**Example 1.** *Suppose that* $\mathcal{X} = \{a, b\}$, *the language of valid strings is* $\mathcal{L} = \{aa, ba\}$, *and the probability distribution is encoded in the following probability tree, which be annotated with the valid and invalid strings:*

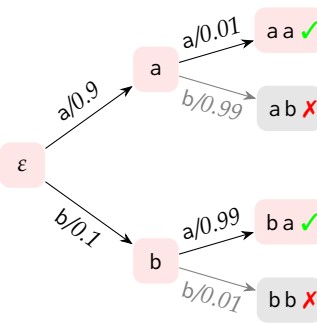

*Under the original distribution, we have*

$p(aa) = .9 \cdot .01 = 0.009$

$p(ab) = .9 \cdot .99 = 0.891$

$p(ba) = .1 \cdot .99 = 0.099$

$p(bb) = .1 \cdot .01 = 0.001$

*The globally conditioned distribution is*

$\vec{P}(aa) = .083333$

$\vec{P}(ba) = .916667$

*This example illustrates a local reversal: under the original distribution* $P_0$, *the first symbol is* a *in .9 of cases, and* b *is the first symbol in .1 cases. However, after conditioning, the case that made* a *so common (i.e.,* ab*) is disallowed. And, less importantly, a minor case for* b *is disallowed. This makes the globally conditioned conditional prefix probability:*

$\vec{P}(a \mid \varepsilon) = \frac{0.009}{0.009 + .099} = .083333$

$\vec{P}(b \mid \varepsilon) = \frac{0.099}{0.009 + .099} = .916667$

*which is a dramatic reversal from .9 and .1, respectively.*

**Importance sampling.** In our setting, importance sampling is a simple sampling-based approximate inference technique that can be used to extend locally constrained data with a weight correction. The weight correction allows us to overcome the bias in $\vec{\ell} \approx \vec{P}$ with additional computation, i.e., taking more samples. Given a computational budget of $M > 0$ samples, the **importance sampling** procedure works as follows:

1. Sample from the locally constrained distribution: $x^{(1)}, \ldots, x^{(M)} \overset{\text{i.i.d.}}{\sim} q$ where $q$ is a **proposal distribution** such as $q = \ell$.

2. Compute weights for each $m$: $w^{(m)} = \frac{P_0(x^{(m)})}{q(x^{(m)})}$. For the special case of $q = \ell$, the weight simplifies to $w^{(m)} = \prod_{t=1}^{|x^{(m)}|+1} L(x_{<t})$.

3. Define estimates :

$$\widehat{G} \stackrel{\text{def}}{=} \frac{1}{M} \sum_{m=1}^{M} w^{(m)} \qquad \widehat{\widetilde{P}}(\pmb{x}) \stackrel{\text{def}}{=} \frac{1}{M} \sum_{m=1}^{M} w^{(m)} \mathbb{1}_{\pmb{x}=\pmb{x}^{(m)}} \qquad \widehat{P}(\pmb{x}) \stackrel{\text{def}}{=} \frac{\widehat{\widetilde{P}}(\pmb{x})}{\widehat{G}}$$

4. Return a sample from the posterior estimate $\widehat{P}$.

**Example 2** (Example 1, continued)**.** *Returning to the above example, in this case, importance sampling with the locally conditioned distribution as proposal generates*

$\ell(\mathsf{a\,a}) = .9$ *with weight* $L(\mathsf{a}) \cdot L(\mathsf{a\,a}) = 1 \cdot .01 = .01$

$\ell(\mathsf{b\,a}) = .1$ *with weight* $L(\mathsf{b}) \cdot L(\mathsf{b\,a}) = 1 \cdot .99 = .99$

*Thus, the importance-weighted distribution estimate is*

$\widehat{P}(\mathsf{a\,a}) = \frac{.01 \cdot .9}{.01 \cdot .9 + .1 \cdot .99} = .083333$

$\widehat{P}(\mathsf{b\,a}) = \frac{.9 + .1}{.01 \cdot .9 + .1 \cdot .99} = .916667$

*which is precisely the global distribution.*

**Sequential Monte Carlo.**   SMC is an extension of importance sampling, which effectively applies importance sampling to a sequence of intermediate target distributions chosen to keep partial generations on track rather than once for the entire sequence.

We define an intermediate **twist function** (also referred to as a **shaping function**) $\vec{\psi}\colon \mathcal{X}^* \to \mathbb{R}_{\geq 0}$, which is the key to providing partially generated strings intermediate feedback. Complete strings, on the other hand, will be judged by $\widetilde{P}$. In this work, we use $\vec{\psi}(\pmb{x}) = \vec{P}_0(\pmb{x})\vec{\mathbb{1}}_{\mathcal{L}}(\pmb{x})$, but we will discuss alternatives shortly. In our incarnation of the SMC algorithm, both complete and incomplete strings will evolve through a time-indexed state space where, at time $t$, the state contains two variables: a string of length $\leq t$, a Boolean indicating if the string is active (incomplete). When the string is active, its length is restricted to equal $t$. Once the string has been completed, it is never changed, i.e., no symbols can be appended to it.

We define the **initial target** as

$$\pi_0(\alpha, \pmb{x}) \stackrel{\text{def}}{=} \mathbb{1}_{\alpha=\top, \pmb{x}=\varepsilon} \tag{12a}$$

which says that initially, the only possible state is $\langle \top, \varepsilon \rangle$, i.e., active and equal to the empty string. We define the **intermediate targets** $\pi_t$ (for $t > 0$) as[11]

$$\begin{aligned} \pi_t(\alpha, \pmb{x}) &= \vec{\psi}(\pmb{x})\mathbb{1}_{|\pmb{x}|=t, \alpha=\top} &\textit{[incomplete]} \\ &+ \widetilde{P}(\pmb{x})\mathbb{1}_{|\pmb{x}|<t, \alpha=\bot} &\textit{[complete; length} < t\textit{]} \end{aligned} \tag{12b}$$

Note that as $t \to \infty$, $\pi_t(\top, \cdot)$ converges to $\widetilde{P}$.

In this work,[12] we use $\vec{\psi}(\pmb{x}) = \vec{P}_0(\pmb{x})\vec{\mathbb{1}}_{\mathcal{L}}(\pmb{x})$ as our shaping function as in (Loula et al., 2025). Under this choice of intermediate target, the constraint checker ensures that we always have at least one valid complete of the string prefix $\pmb{x}$. This feedback is useful for detecting rejection as soon as the string is guaranteed to fail, and it is useful for detecting cases where a lot of the probability mass from the prior has been eliminated (i.e., we are likely to have a low-weight particle in the sense of importance sampling). In relation to $\vec{P}$, it provides a

---

[11]A complete string $\pmb{x}$ requires $|\pmb{x}| + 1$ steps to be generated due to the final $\square$ event. This is why $\pi_t$ has complete strings of length $< t$, rather than (say) $\leq t$.

[12]A guiding principle for designing $\vec{\psi}$ is to approximate $\vec{\psi} \approx \vec{P}$. However, any choice of $\vec{\psi}$ that satisfies the technical condition $\vec{\psi}(\pmb{x}) = 0 \implies \vec{P}(\pmb{x}) = 0$ will converge (albeit with different rates). Note that if $\vec{\psi} = \vec{P}$, the SMC algorithm is an *exact* sampler for $P$. Unfortunately, computing $\vec{P}$ exactly is typically intractable, so we must approximate it.

sufficient condition for when $\vec{P}(x) = 0$. It, however, does not provide much information beyond that.[13]

We define the shorthand

$$\vec{\psi}(x' \mid x) \stackrel{\text{def}}{=} \begin{cases} \dfrac{\widetilde{P}(x)}{\overline{\psi}(x)} & \textbf{if } x' = \square \\[2ex] \dfrac{\overline{\psi}(x\,x')}{\overline{\psi}(x)} & \textbf{otherwise} \end{cases} \tag{13}$$

Importantly, this definition gives the following factorization of $\widetilde{P}$, $\forall x \in \mathcal{X}^*$,

$$\widetilde{P}(x) = \vec{\psi}(\varepsilon)\vec{\psi}(\square \mid x)\prod_{t=1}^{|x|}\vec{\psi}(x_t \mid x_{<t}) \tag{14}$$

It is straightforward to verify that the shaped weights of complete strings are equivalent to those used in importance sampling:

$$\frac{\widetilde{P}(x)}{q(x)} = \vec{\psi}(\varepsilon)\frac{\vec{\psi}(\square \mid x)}{\vec{q}(\square \mid x)}\prod_{t=1}^{|x|}\frac{\vec{\psi}(x_t \mid x_{<t})}{\vec{q}(x_t \mid x_{<t})} \tag{15}$$

We provide pseudocode for the SMC procedure in Alg. 1.

---

**Algorithm 1** Sequential Monte Carlo

1: **procedure** SMC($q, \vec{\psi}, M, \tau$)
2:    **for** $m = 1 \dots M$ :          ▷$M$ number of samples
3:      $(x^{(m)}, w^{(m)}, \alpha^{(m)}) \leftarrow (\varepsilon, \vec{\psi}(\varepsilon), \texttt{true})$
4:    **while** $\exists m \in 1 \dots M \colon \alpha^{(m)}$ :
5:      **for** $m = 1 \dots M$ s.t. $\alpha^{(m)}$ :          ▷Incomplete particles
6:        $x' \sim q(\cdot \mid x^{(m)})$
7:        **if** $x' = \square$ :          ▷Complete particle
8:          $\alpha^{(m)} \leftarrow \texttt{false}$
9:        **else**
10:          $x^{(m)} \leftarrow x^{(m)} \circ x'$
11:          $w^{(m)} \leftarrow w^{(m)} \dfrac{\vec{\psi}(x' \mid x^{(m)})}{q(x' \mid x^{(m)})}$
12:      $(x^{(\cdot)}, w^{(\cdot)}, \alpha^{(\cdot)}) \leftarrow \textsc{Resample}(x^{(\cdot)}, w^{(\cdot)}, \alpha^{(\cdot)}, \tau)$
13:    $\widehat{G} \leftarrow \frac{1}{M}\sum_{m=1}^{M} w^{(m)}$
14:    $\widehat{\widetilde{P}}(x) \leftarrow \frac{1}{M}\sum_{m=1}^{M} w^{(m)}\mathbb{1}\{x = x^{(m)}\}$
15:    $\widehat{P}(x) \leftarrow \dfrac{\widehat{\widetilde{P}}(x)}{\widehat{G}}$
16:    **return** $(\widehat{G}, \widehat{\widetilde{P}}, \widehat{P})$

17: **procedure** RESAMPLE($x^{(\cdot)}, w^{(\cdot)}, \alpha^{(\cdot)}, \tau$)
18:    $W \leftarrow \sum_{m=1}^{M} w^{(m)}$
19:    $\widehat{M} \leftarrow W^2 / \left(\sum_{m=1}^{M}\left(w^{(m)}\right)^2\right)$          ▷Effective sample size
20:    **if** $\widehat{M} < \tau \cdot M$ :          ▷Resample if needed
21:      $\overline{x}^{(\cdot)} \leftarrow x^{(\cdot)}$; $\overline{w}^{(\cdot)} \leftarrow w^{(\cdot)}$          ▷Temporary copy
22:      **for** $m = 1 \dots M$ :
23:        $R \sim \text{Categorical}(\frac{1}{W}\langle \overline{w}^{(1)}, \dots, \overline{w}^{(M)}\rangle)$
24:        $(x^{(m)}, w^{(m)}, \alpha^{(m)}) \leftarrow (\overline{x}^{(R)}, W/M, \alpha^{(R)})$
25:    **return** $(x^{(\cdot)}, w^{(\cdot)}, \alpha^{(\cdot)})$

---

[13]Note that it is ok from the perspective of the technical conditions for $\vec{\mathbb{1}}_{\mathcal{L}}(x)$ to have false positives (i.e., strings that are considered ok, that are not); however, false negatives are prohibited.

**SMC extension for properly weighted token proposals.** Our method uses an extension of Alg. 1 that allows for a properly weighted proposal distribution (See App. B for a more detailed discussion of proper weighting). More specifically, the token proposals will now be a pair of a token and a weight, i.e., $(x', w') \sim q(\cdot \mid \boldsymbol{x}^{(m)})$. The requirement is that token proposal distribution is properly weighted with respect to the intermediate target: $\vec{\psi}(x' \mid \boldsymbol{x}^{(m)})$. While still yielding sound inference, the use of approximate proposals does admit additional variance. We provide several methods for defining these kinds of proposal distributions (§3). Pseudocode for the extended method is provided in Alg. 2.

---

**Algorithm 2** Sequential Monte Carlo with Properly Weighted Proposal

---

1. **procedure** SMC-PWP$(q, \vec{\psi}, M, \tau)$
2.   **for** $m = 1 \dots M$ : $\qquad\qquad\qquad\qquad\qquad\qquad\qquad\qquad\qquad$ ▷$M$ *number of samples*
3.     $(\boldsymbol{x}^{(m)}, w^{(m)}, \alpha^{(m)}) \leftarrow (\varepsilon, \vec{\psi}(\varepsilon), \texttt{true})$
4.   **while** $\exists m \in 1 \dots M : \alpha^{(m)}$ :
5.     **for** $m = 1 \dots M$ s.t. $\alpha^{(m)}$ : $\qquad\qquad\qquad\qquad\qquad\qquad$ ▷*Incomplete particles*
6.       $(x', w') \sim q(\cdot \mid \boldsymbol{x}^{(m)})$
7.       **if** $x' = \square$ : $\qquad\qquad\qquad\qquad\qquad\qquad\qquad\qquad\qquad$ ▷*Complete particle*
8.         $\alpha^{(m)} \leftarrow \texttt{false}$
9.       **else**
10.         $\boldsymbol{x}^{(m)} \leftarrow \boldsymbol{x}^{(m)} \circ x'$
11.       $w^{(m)} \leftarrow w^{(m)} \cdot w'$
12.     $(\boldsymbol{x}^{(\cdot)}, w^{(\cdot)}, \alpha^{(\cdot)}) \leftarrow \textsc{Resample}(\boldsymbol{x}^{(\cdot)}, w^{(\cdot)}, \alpha^{(\cdot)}, \tau)$
13.   $\widehat{G} \leftarrow \frac{1}{M} \sum_{m=1}^{M} w^{(m)}$
14.   $\widehat{\widetilde{P}}(\boldsymbol{x}) \leftarrow \frac{1}{M} \sum_{m=1}^{M} w^{(m)} \mathbb{1}\{\boldsymbol{x} = \boldsymbol{x}^{(m)}\}$
15.   $\widehat{P}(\boldsymbol{x}) \leftarrow \frac{\widehat{\widetilde{P}}(\boldsymbol{x})}{\widehat{G}}$
16.   **return** $(\widehat{G}, \widehat{\widetilde{P}}, \widehat{P})$

---

# B   Recursive Auxiliary-Variable Inference (RAVI)

**Importance sampling** is an approach to approximate sampling from an unnormalized target distribution $\widetilde{p}$, with normalizing constant $Z$, through a proposal distribution $q$ whose support is at least that of $\widetilde{p}$. We would like to design $q$ with favorable properties towards our downstream goal, e.g., efficient runtime. Crucially, this design of a strong proposal distribution $q$ is often mediated by auxiliary random choices $r$ such that we sample $\langle x, r \rangle \sim q$. When this is the case, it is necessary to calculate a weight $w(x, r)$ to correct for $r$.

RAVI ([Lew et al., 2022](#)) gives us a generalized, flexible recipe for deriving the weights of relatively arbitrary choices of $q$. Within RAVI, we may work with any unnormalized target distribution $\widetilde{p}$ over any space $\mathcal{X}$. Let $\langle x, w \rangle \sim Q$ denote the process of sampling from a proposal $q$ and calculating its weight $w$. We are interested in developing $Q$ with the following property:

**Definition 3.** *The proposal distribution $Q$ is **properly weighted** if:*

$$\mathop{\mathbb{E}}_{\langle x,w \rangle \sim Q} [w\, f(x)] = Z \mathop{\mathbb{E}}_{x \sim p} [f(x)] \tag{16}$$

Note that by taking trivial $f(x) = 1$, this property implies $\mathbb{E}_{\langle x,w \rangle \sim Q}[w] = Z$. Leveraging this identity, each estimator $\widehat{Z}$ in the main text is approximated via the weight $w$ of a properly weighted proposal distribution. For notational consistency with the literature on proper weighting, $w$ will be used as opposed to $\widehat{Z}$ throughout the appendix.

## B.1   The Proposal

A RAVI proposal is a joint distribution $q(r, x)$ over a product space $\mathcal{R} \times \mathcal{X}$, where $\mathcal{R}$ holds auxiliary random choices made by the proposal. Typically, proposals will be designed to make the marginal $q(x)$ a good approximation to the target $p(x)$.

In this work, we consider exclusively exact proposals, where $q(x) = p(x)$. This is achieved via various formulations of rejection sampling, where $r$ represents the auxiliary randomness generated by the rejection sampler. Because $q(x) = \sum_{r \in \mathcal{R}} q(r, x)$ is typically intractable to evaluate exactly, the usual importance weight $w = \frac{\widetilde{p}(x)}{q(x)}$ cannot be computed directly. RAVI provides a way to tractably compute a (noisy) importance weight that still satisfies proper weighting.

## B.2   The Meta-Proposal

A RAVI meta-proposal $h(r; x)$ is designed to infer $r$ given $x$. The optimal meta-proposal would be $q(r|x) \stackrel{\text{def}}{=} q(r, x)/q(x)$, but this optimal choice is often not tractable. In practice, we select a family of probability distributions $h(r; x)$ over $\mathcal{R}$, indexed by $x \in \mathcal{X}$, with the appropriate support, in order to define an 'extended' target $(\widetilde{p}h)(r, x) \stackrel{\text{def}}{=} \widetilde{p}(x)h(r; x)$ over the joint space. Formally, we require that $\widetilde{p}h \ll q$.[14] For a given $x$, this means that $h$ should, with probability 1, propose some $r$ such that $q(r, x) > 0$.

## B.3   Properly Weighted Sampling

**Definition 4.** *We define **1-level RAVI sampling** (cf. 2-level RAVI sampling in Def. [5]) from a proposal $q$ and meta-proposal $h$ as follows:*

1. *Generate $(r, x) \sim q$.*
2. *Evaluate $w \stackrel{\text{def}}{=} \frac{\widetilde{p}(x)\, h(r; x)}{q(r, x)}$.*
3. *Return $\langle x, w \rangle$.*

---

[14]For $\mu, \nu$ two distributions on the same domain, $\mu$ is said to be **absolutely continuous** (AC) with respect to $\nu$ (written $\mu \ll \nu$) if and only if $\mu$ is zero anywhere that $\nu$ is zero.

This can be seen as standard importance sampling on the extended state space $\mathcal{R} \times \mathcal{X}$. The weighted value $\langle\langle r, x \rangle, w\rangle$ is properly weighted for the extended target $\widetilde{p}(x)\, h(r; x)$, which means that the weighted value $\langle x, w\rangle$ is properly weighted for the re-marginalized target $\widetilde{p}(x)$. When $h(r; x)$ performs "perfect meta-inference," i.e., when $h(r; x) = q(r|x)$, then $w = \frac{\widetilde{p}(x)\, h(r;x)}{q(r,x)} = \frac{\widetilde{p}(x)\, q(r|x)}{q(r|x)\, q(x)} = \frac{\widetilde{p}(x)}{q(x)}$, meaning we compute exact importance weights. Otherwise, the importance weights will be noisier but will still have the same expected value (that is, $\mathbb{E}[w \mid x] = \frac{\widetilde{p}(x)}{q(x)}$; consequently we also have $\mathbb{E}[w] = \mathbb{E}_{(r,x)\sim q}[\frac{\widetilde{p}(x)}{q(x)}] = Z$).

**Proposition 5.** *Given a proposal $q(r, x)$ and meta-proposal $h(r; x)$, such that $\widetilde{p}\, h \ll q$, Def. 4 is properly weighted for $\widetilde{p}$.*

*Proof.*

$$
\begin{aligned}
\mathbb{E}_{\langle x,w\rangle \sim Q_{\text{1-RAVI}}} [w\, f(x)] &= \mathbb{E}_{\langle r,x\rangle \sim q} \left[ \frac{\widetilde{p}(x)\, h(r; x)}{q(r, x)}\, f(x) \right] && \textit{[Def. 4]} && \text{(17a)} \\
&= \sum_{r \in \mathcal{R}} \sum_{x \in \mathcal{X}} q(r, x)\, \frac{\widetilde{p}(x)\, h(r; x)}{q(r, x)}\, f(x) && \textit{[Def. of expectation]} && \text{(17b)} \\
&= \sum_{r \in \mathcal{R}} \sum_{x \in \mathcal{X}} \widetilde{p}(x)\, h(r; x)\, f(x) && \textit{[Cancellation; AC]} && \text{(17c)} \\
&= \sum_{x \in \mathcal{X}} \widetilde{p}(x)\, f(x) \sum_{r \in \mathcal{R}} h(r; x) && \textit{[Rearranging]} && \text{(17d)} \\
&= \sum_{x \in \mathcal{X}} \widetilde{p}(x)\, f(x) && \textit{[$h(\cdot; x)$ is a distribution]} && \text{(17e)} \\
&= \sum_{x \in \mathcal{X}} Z\, p(x)\, f(x) && \textit{[Def. of $p$]} && \text{(17f)} \\
&= Z \sum_{x \in \mathcal{X}} p(x)\, f(x) && \textit{[Rearranging]} && \text{(17g)} \\
&= Z\, \mathbb{E}_{x \sim p}[f(x)] && \textit{[Def. of expectation]} && \text{(17h)}
\end{aligned}
$$

$\blacksquare$

### B.4 Deeper Recursive Inference

If our meta-inference $h$ itself introduces auxiliary variables $s \in \mathcal{S}$ (so we have $h(s, r; x)$), then we can repeat this process, adding a meta-meta-proposal $j(s; r, x)$ that aims to approximate $h(s \mid r; x)$. Our properly weighted algorithm then becomes:

**Definition 5.** *We define 2-level RAVI sampling from a proposal $q$, meta-proposal $h$, and meta-meta-proposal $j$ as follows:*

1. *Generate $(r, x) \sim q$ and $s \sim j(\cdot; r, x)$.*
2. *Evaluate $w \overset{\text{def}}{=} \frac{\widetilde{p}(x)\, h(s, r; x)}{q(r, x)\, j(s; r, x)}$.*
3. *Return $\langle x, w\rangle$.*

Intuitively, since $h$ "overextended" the target distribution to include not just $r$ but also $s$, we must now extend the proposal $q$ with a distribution $j$ over $s$. In general, we can continue this process (e.g., if $j$ has auxiliary variables), alternately extending the model and proposal until we reach an extension that does not introduce new auxiliary variables.

**Proposition 6.** *Given a proposal $q(r, x)$, meta-proposal $h(s, r; x)$, and meta-meta-proposal $j(s; r, x)$, such that $\widetilde{p}\, h \ll q\, j$, Def. 5 is properly weighted for $\widetilde{p}$.*

*Proof.*

$$\mathbb{E}_{\langle x,w\rangle \sim Q_{\text{2-RAVI}}} [w\, f(x)]$$

$$= \mathbb{E}_{\langle r,x\rangle \sim q, s\sim j(\cdot;r,x)} \left[ \frac{\widetilde{p}(x)\, h(s,r;x)}{q(r,x)\, j(s;r,x)}\, f(x) \right] \qquad\qquad \textit{[Def. 5]} \qquad (18a)$$

$$= \sum_{r\in\mathcal{R}} \sum_{x\in\mathcal{X}} \sum_{s\in\mathcal{S}} q(r,x)\, j(s;r,x)\, \frac{\widetilde{p}(x)\, h(s,r;x)}{q(r,x)\, j(s;r,x)}\, f(x) \qquad \textit{[Def. of expectation]} \qquad (18b)$$

$$= \sum_{r\in\mathcal{R}} \sum_{x\in\mathcal{X}} \sum_{s\in\mathcal{S}} \widetilde{p}(x)\, h(s,r;x)\, f(x) \qquad\qquad \textit{[Cancellation; AC]} \qquad (18c)$$

$$= \sum_{x\in\mathcal{X}} \widetilde{p}(x)\, f(x) \sum_{r\in\mathcal{R}} \sum_{s\in\mathcal{S}} h(s,r;x) \qquad\qquad \textit{[Rearranging]} \qquad (18d)$$

$$= \sum_{x\in\mathcal{X}} \widetilde{p}(x)\, f(x) \sum_{r\in\mathcal{R}} h(r;x) \qquad\qquad \textit{[Marginalize over s]} \qquad (18e)$$

$$= \sum_{x\in\mathcal{X}} \widetilde{p}(x)\, f(x) \qquad\qquad \textit{[$h(\cdot;x)$ is a distribution]} \qquad (18f)$$

$$= \sum_{x\in\mathcal{X}} Z\, p(x)\, f(x) \qquad\qquad \textit{[Def. of $p$]} \qquad (18g)$$

$$= Z \sum_{x\in\mathcal{X}} p(x)\, f(x) \qquad\qquad \textit{[Rearranging]} \qquad (18h)$$

$$= Z\, \mathbb{E}_{x\sim p} [f(x)] \qquad\qquad \textit{[Def. of expectation]} \qquad (18i)$$

∎

### B.5 RAVI Intuitions for Def. 1 (WRS)

In the context of the RAVI framework, WRS can be understood as representing the rejected samples as auxiliary variables generated during the sampling process. The auxiliary space $\mathcal{R} = \cup_{i\in\mathbb{N}}\mathcal{X}^i$ consists of finite lists of rejected samples. The proposal $q$ generates a trace of rejection sampling, while the meta-proposal $h$ generates $L$ additional rejection loops to improve our inference of the auxiliary variables. Finally, a meta-meta-proposal $j$ accounts for the additional auxiliary variables introduced by $h$.

Intuitively, as $L$ increases, we obtain a better estimate of the acceptance probability $Z$, leading to lower variance in our importance weights. In the limit as $L \to \infty$, meta-proposal $h(r;x)$ converges to its optimal distribution $q(r \mid x)$ and the meta-meta-proposal $j(s;r,x)$ converges to its optimal distribution $h(s \mid r;x)$.

### B.6 RAVI Intuitions for Def. 2 (AWRS)

In the context of the RAVI framework, AWRS represents a modification where the sampling procedures in the proposal, meta-proposal, and meta-meta-proposal all maintain memory of previously rejected samples. The sampling distributions are continually renormalized after each rejection to account for the removed probability mass. This adaptation requires a different weight calculation that accounts for the changing probability distributions throughout the sampling process.

## C   Proofs for §3.1 (WRS)

**Proposition 1.** *For $\langle x, \widehat{Z} \rangle \sim Q_{WRS}$, $x$ is distributed according to $p$ and $\mathbb{E}[\widehat{Z}] = Z$.*      *(App. C)*

Recall $\widehat{Z} = w$, where $w$ is the weight of a properly weighted proposal distribution. $w$ will be used in this section for consistency with notational convention from proper weighting (see App. B for discussion).

First, we will show that $x$ is distributed according to $p$.

*Proof.*

$$\Pr_{\langle x,w \rangle \sim Q_{WRS}}[x] = \Pr_{x \sim p_0}[x \mid \mathbb{1}_{\mathcal{C}}(x)] \hspace{3cm} \textit{[Def. 1]} \hspace{1cm} \text{(19a)}$$

$$= \frac{\Pr_{x \sim p_0}[x] \Pr_{x \sim p_0}[\mathbb{1}_{\mathcal{C}}(x) \mid x]}{\Pr_{x \sim p_0}[\mathbb{1}_{\mathcal{C}}(x)]} \hspace{1.5cm} \textit{[Bayes rule]} \hspace{0.5cm} \text{(19b)}$$

$$= \frac{p_0(x) \mathbb{1}_{\mathcal{C}}(x)}{Z} \hspace{2.5cm} \textit{[Defs. of $p_0$, $\mathbb{1}_{\mathcal{C}}$, $Z$]} \hspace{0.3cm} \text{(19c)}$$

$$= p(x) \hspace{4cm} \textit{[Def. of $p$]} \hspace{0.7cm} \text{(19d)}$$

$\blacksquare$

Next, we will prove the unbiasedness of $\widehat{Z} = w$ as an estimator for $Z$ by proving that $Q_{WRS}$ is properly weighted for $\widetilde{p}$ (Def. 3).

*Proof.* We will prove this using the 2-level RAVI framework from Def. 5.

First, we define our spaces and distributions:

- Let the auxiliary space $\mathcal{R} = \cup_{i \in \mathbb{N}} \mathcal{X}^i$ represent finite lists of samples rejected before obtaining an accepted sample.
- For $r = (r_1, \dots, r_n) \in \mathcal{R}$ and $x \in \mathcal{X}$, define our proposal distribution (Remember $\mathbb{1}_{\mathcal{C}} : \mathcal{X} \to \{0,1\}$):

$$q(r,x) = p_0(x_c) \mathbb{1}_{\mathcal{C}}(x_c) \prod_{i=1}^{n} p_0(r_i) (1 - \mathbb{1}_{\mathcal{C}}(r_i)) \hspace{1.5cm} \textit{[Def.]} \hspace{0.7cm} \text{(20a)}$$

$$= p_0(x_c) \mathbb{1}_{\mathcal{C}}(x_c) \prod_{i=1}^{n} p_0(r_i) \hspace{1.5cm} \textit{[$\forall r \in \mathcal{R}: \mathbb{1}_{\mathcal{C}}(r) = 0$]} \hspace{0.3cm} \text{(20b)}$$

$$= p_0(x_c) \prod_{i=1}^{n} p_0(r_i) \hspace{1.7cm} \textit{[$\forall x_c \in \mathcal{C}: \mathbb{1}_{\mathcal{C}}(x_c) = 1$]} \hspace{0.3cm} \text{(20c)}$$

This represents the probability of rejecting $r_1 \cdots r_n$ and accepting $x_c$.
- Our meta-proposal $h$ requires additional auxiliary variables. Let $\mathcal{S} = (\mathcal{R} \times \mathcal{X})^L \times \{1, \dots, L\}$, representing $L$ additional rejection loops along with an index $L^*$ choosing one loop. Recall $h$ aims to infer the rejected $r$ given $x$. But since $x$ is independent of the rejected samples that preceded it, we set $h$ to simulate its own $L$ rejection sampling loop(s). To avoid this same problem at the next level of meta-inference, we leave auxiliary randomness consisting of sequences that can be simulated by rejection sampling in $j$ below.
- For $s = ((s^{(1)}, \dots, s^{(L)}), (s_1^*, \dots, s_L^*), L^*) \in \mathcal{S}, r \in \mathcal{R}, x \in \mathcal{X}$, set the meta-proposal to generate $L$ rejection loops where each $s^{(i)}$ represents rejected samples, $s_i^*$ is an accepted sample, and $L^*$ is the index of the split loop,[15] which is chosen with probability proportional to $n_i + 1$, where $n_i = |s^{(i)}|$. Return the chosen $s^{(L^*)}$ truncated at the sampled split-point as

---

[15] Note, the terms containing $L^*$ will cancel out, so we do not actually need to sample it.

the proposed latent sequence of rejected samples. Letting $n_0 = n$ for convenience:

$$
h(s, r; x)
$$

$$
= \frac{n_0 + n_{L^*} + 1}{\sum_{i=0}^{L}(n_i + 1)} \frac{1}{n_0 + n_{L^*} + 1}
$$

$$
\cdot \prod_{i=1}^{L} \left( p_0(s_i^*) \, \mathbb{1}_\mathcal{C}(s_i^*) \prod_{j=1}^{n_i} p_0(s_j^{(i)})(1 - \mathbb{1}_\mathcal{C}(s_j^{(i)})) \right) \qquad \text{[Def.]} \quad (21a)
$$

$$
= \frac{1}{\sum_{i=0}^{L}(n_i + 1)} \prod_{i=1}^{L} \left( p_0(s_i^*) \, \mathbb{1}_\mathcal{C}(s_i^*) \prod_{j=1}^{n_i} p_0(s_j^{(i)})(1 - \mathbb{1}_\mathcal{C}(s_j^{(i)})) \right) \quad \text{[Cancellation]} \quad (21b)
$$

$$
= \frac{1}{\sum_{i=0}^{L}(n_i + 1)} \prod_{i=1}^{L} \left( p_0(s_i^*) \, \mathbb{1}_\mathcal{C}(s_i^*) \prod_{j=1}^{n_i} p_0(s_j^{(i)}) \right) \quad \text{[}\forall s^{(i)} \in \mathcal{S} : \mathbb{1}_\mathcal{C}\left(s^{(i)}\right) = 0\text{]} \quad (21c)
$$

$$
= \frac{1}{\sum_{i=0}^{L}(n_i + 1)} \prod_{i=1}^{L} \left( p_0(s_i^*) \prod_{j=1}^{n_i} p_0(s_j^{(i)}) \right) \quad \text{[}\forall s^* \in \mathcal{S} : \mathbb{1}_\mathcal{C}(s^*) = 1\text{]} \quad (21d)
$$

- Our meta-meta-proposal $j$ aims to infer the auxiliary randomness $s$ given $r$. Again, $r$ is not useful, but the sequence of rejected samples in $s$ can be simulated by rejection sampling. Set $j$ to generate $L$ independent rejection loops and select $L^*$ uniformly, representing the guess about which loop was split. Note that due to independence, we can re-use our existing loops. $j$ is defined as follows:

$$
j(s; r, x) = \frac{1}{L} \prod_{i=1}^{L} \left( p_0(s_i^*) \, \mathbb{1}_\mathcal{C}(s_i^*) \prod_{j=1}^{n_i} p_0(s_j^{(i)})(1 - \mathbb{1}_\mathcal{C}(s_j^{(i)})) \right) \quad \text{[Def.]} \quad (22a)
$$

$$
= \frac{1}{L} \prod_{i=1}^{L} \left( p_0(s_i^*) \, \mathbb{1}_\mathcal{C}(s_i^*) \prod_{j=1}^{n_i} p_0(s_j^{(i)}) \right) \quad \text{[}\forall s^{(i)} \in \mathcal{S} : \mathbb{1}_\mathcal{C}\left(s^{(i)}\right) = 0\text{]} \quad (22b)
$$

$$
= \frac{1}{L} \prod_{i=1}^{L} \left( p_0(s_i^*) \prod_{j=1}^{n_i} p_0(s_j^{(i)}) \right) \quad \text{[}\forall s^* \in \mathcal{S} : \mathbb{1}_\mathcal{C}(s^*) = 1\text{]} \quad (22c)
$$

Following Def. 5, and substituting our definitions, we calculate the proper weight as follows. Note all terms of $p_0$ and $\mathbb{1}_\mathcal{C}$ will cancel out:

$$
w = \frac{\widetilde{p}(x) h(s, r; x)}{q(r, x) j(s; r, x)} \qquad \text{[Def. 5]} \quad (23a)
$$

$$
= \frac{p_0(x) \mathbb{1}_\mathcal{C}(x) \frac{1}{\sum_{i=0}^{L}(n_i+1)} \prod_{i=1}^{L}(\ldots)}{p_0(x_c) \prod_{i=1}^{n} p_0(r_i) \frac{1}{L} \prod_{i=1}^{L}(\ldots)} \qquad \text{[Substitution of defs.]} \quad (23b)
$$

$$
= \frac{\frac{1}{\sum_{i=0}^{L}(n_i+1)}}{\frac{1}{L}} \qquad \text{[Cancellation; AC]} \quad (23c)
$$

$$
= \frac{L}{\sum_{i=0}^{L}(n_i + 1)} \qquad \text{[Algebra]} \quad (23d)
$$

$$
= \frac{L}{L + \sum_{i=0}^{L} n_i} \qquad \text{[Algebra]} \quad (23e)
$$

$$
= \frac{L}{L + n} \qquad \text{[Def. of } n\text{]} \quad (23f)
$$

This matches the weight formula in Def. 1, confirming that our sampling procedure is properly weighted for $\widetilde{p}$ justified by the RAVI framework. ∎

### C.1 Another perspective on WRS

Upon completing the RAVI derivation, it is now clear that an alternative formulation for the weights of WRS is possible. We can consider our weight calculation as using $L + 1$ independent rejection-sampling loops to estimate the acceptance probability $Z$. The total number of trials $T \stackrel{\text{def}}{=} \sum_{i=0}^{L}(n+1)$ required to reach $b \stackrel{\text{def}}{=} L+1$ acceptances in such a process follows a Negative Binomial distribution, i.e., $T \sim NB(b, Z)$. Thus, calculation of $w$ for WRS amounts to calculating an unbiased estimator of $Z$ for the Negative Binomial distribution. See §3.1 for another gloss of this perspective.

**Proposition 7.** $\widehat{Z} = \frac{L}{L+n}$ *is the minimum-variance unbiased estimator (MVUE) for Z.*

*Proof.* The PMF of $NB(b, Z)$, for $b \in \mathbb{N}_{\geq 0}, Z \in [0, 1]$ is:

$$\Pr[T = t; b, Z] = \binom{t-1}{b-1} Z^b (1-Z)^{t-b} \tag{24}$$

supported on $t \in \{b, b+1, b+2, \dots\}$.

Assuming $b = L + 1$ is fixed, the PMF may be written in exponential family form:

$$\Pr[T = t; b, Z] = \binom{t-1}{b-1} \exp\left(b \log Z + (t-b) \log(1-Z)\right) \tag{25a}$$

$$= \binom{t-1}{b-1} \exp\left(t \log(1-Z) + b \log\left(\frac{Z}{1-Z}\right)\right) \tag{25b}$$

with sufficient statistic $t$. Since $t = b + n$, $b$ is fixed, and we consider a one-parameter exponential family, we can conclude that $n$ is a complete sufficient statistic for $Z$.

Let's start with a simple unbiased estimator for $Z$. Consider the indicator random variable $\mathbb{1}_{x_0 \in \mathcal{C}}$, denoting the success of the first trial. Trivially, $\mathbb{E}[\mathbb{1}_{x_0 \in \mathcal{C}}] = \Pr[x_0 \in \mathcal{C}] = Z$. By the Rao-Blackwell theorem, we can reduce the variance of this estimator by computing its conditional expectation given the sufficient statistic $n$:

$$\widehat{Z} = \mathbb{E}[\mathbb{1}_{x_0 \in \mathcal{C}} \mid n] \tag{26a}$$

$$= \Pr[x_0 \in \mathcal{C} \mid n] \qquad\qquad \textit{[Def. of Expectation]} \tag{26b}$$

$$= \frac{\Pr[n \mid x_0 \in \mathcal{C}] \Pr[x_0 \in \mathcal{C}]}{\Pr[n]} \qquad\qquad \textit{[Bayes Rule]} \tag{26c}$$

$$= \frac{\binom{(b-1)+n-1}{(b-1)-1} Z^{(b-1)} (1-Z)^n \cdot Z}{\binom{b+n-1}{b-1} Z^b (1-Z)^n} \qquad\qquad \textit{[Substitute Defs.]} \tag{26d}$$

$$= \frac{\binom{(b-1)+n-1}{(b-1)-1} Z^b (1-Z)^n}{\binom{b+n-1}{b-1} Z^b (1-Z)^n} \qquad\qquad \textit{[Combine like terms]} \tag{26e}$$

$$= \frac{\binom{(b-1)+n-1}{(b-1)-1}}{\binom{b+n-1}{b-1}} \qquad\qquad \textit{[Cancellation]} \tag{26f}$$

$$= \frac{b-1}{b+n-1} \qquad\qquad \textit{[Algebra]} \tag{26g}$$

$$= \frac{L+1-1}{L+1+n-1} \qquad\qquad \textit{[b=L+1]} \tag{26h}$$

$$= \frac{L}{L+n} \qquad\qquad \textit{[Algebra]} \tag{26i}$$

So, $\widehat{Z} = \frac{L}{L+n}$ is our estimator.

To confirm that $\widehat{Z}$ is unbiased, see that its expected value reduces to summing over the entire support of the PMF of $T' \sim NB(b-1, Z)$, scaled by constant $Z$.

$$\mathbb{E}[\widehat{Z}] = \mathbb{E}\left[\frac{L}{L+n}\right] \tag{27a}$$

$$= \sum_{t=b}^{\infty}\left[\frac{L}{L+n}\Pr[T=t;b,Z]\right] \quad\quad \textit{[Def. of Expectation]} \tag{27b}$$

$$= \sum_{t=b}^{\infty}\left[\frac{b-1}{t-1}\binom{t-1}{b-1}Z^b(1-Z)^{t-b}\right] \quad\quad \textit{[Substitute Defs.]} \tag{27c}$$

$$= \sum_{t=b}^{\infty}\left[\binom{t-2}{b-2}Z^b(1-Z)^{t-b}\right] \quad\quad \textit{[Binom. Coef. Identity]} \tag{27d}$$

$$= \sum_{t'=b-1}^{\infty}\left[\binom{t'-1}{b-2}Z^b(1-Z)^{t'-(b-1)}\right] \quad\quad \textit{[Reindex s.t. } t' = t-1] \tag{27e}$$

$$= Z\sum_{t'=b-1}^{\infty}\left[\binom{t'-1}{(b-1)-1}Z^{b-1}(1-Z)^{t'-(b-1)}\right] \quad\quad \textit{[Algebra]} \tag{27f}$$

$$= Z\sum_{t'=b-1}^{\infty}\Pr[T'=t';b-1,Z] \quad\quad \textit{[Eq. (24)]} \tag{27g}$$

$$= Z \quad\quad \textit{[Dist. Sums to 1]} \tag{27h}$$

Therefore $\widehat{Z}$ is an unbiased estimator of $Z$. Note this proof relies on $b = L + 1 \geq 2$, which we have by construction, since $L \geq 1$.

By the Lehmann-Scheffé theorem, since $n$ is a complete sufficient statistic for $Z$, and $\widehat{Z}$ is an unbiased estimator that is a function of $n$, $\widehat{Z}$ must be the minimum-variance unbiased estimator (MVUE) for $Z$. ∎

# D Proofs for §3.2 (AWRS)

**Proposition 3.** *For $\langle x, \widehat{Z} \rangle \sim Q_{AWRS}$, $x$ is distributed according to $p$ and $\mathbb{E}[\widehat{Z}] = Z$.* *(App. D)*

As in App. C, $\widehat{Z} = w$, and $w$ will be used for consistency with notational convention on proper weighting (see App. B for discussion).

The first statement, that $x$ is distributed according to $p$, follows directly from App. C and Def. 2, i.e., that rejection samplers yield exact samples and AWRS is a rejection sampler.

Next, to prove the unbiasedness of $\widehat{Z} = w$ as an estimator for $Z$, we will prove that $Q_{AWRS}$ is properly weighted for $\widetilde{p}$ (Def. 3).

*Proof.* We will prove this using the 2-level RAVI framework from Def. 5, adapted for the case where our sampling procedures are adaptive.

First, we define our spaces and distributions with adaptivity:

- Let the auxiliary space $\mathcal{R} = \cup_{i \in \mathbb{N}} \mathcal{X}^i$ represent finite lists of distinct samples rejected before obtaining an accepted sample.
- For $r = (r_1, \dots, r_{n_0}) \in \mathcal{R}$ and $x \in \mathcal{X}$, we define our adaptive proposal distribution:

$$q(r,x)$$
$$= \left( \prod_{i=1}^{n_0} \frac{1}{1 - \sum_{j=1}^{i-1} p_0(r)} \right) p_0(x_c) \, \mathbb{1}_{\mathcal{C}}(x_c) \prod_{i=1}^{n_0} p_0(r_i)(1 - \mathbb{1}_{\mathcal{C}}(r_i)) \, \mathbb{1}_{[\forall i \neq j. r_i \neq r_j]} \qquad [Def.] \tag{28a}$$

$$= \left( \prod_{i=1}^{n_0} \frac{1}{1 - \sum_{j=1}^{i-1} p_0(r)} \right) p_0(x_c) \, \mathbb{1}_{\mathcal{C}}(x_c) \prod_{i=1}^{n_0} p_0(r_i) \, \mathbb{1}_{[\forall i \neq j. r_i \neq r_j]} \qquad [\forall r \in \mathcal{R} : \mathbb{1}_{\mathcal{C}}(r) = 0] \tag{28b}$$

$$= \left( \prod_{i=1}^{n_0} \frac{1}{1 - \sum_{j=1}^{i-1} p_0(r)} \right) p_0(x_c) \prod_{i=1}^{n_0} p_0(r_i) \, \mathbb{1}_{[\forall i \neq j. r_i \neq r_j]} \qquad [\forall x_c \in \mathcal{C} : \mathbb{1}_{\mathcal{C}}(x) = 1] \tag{28c}$$

This represents the probability of rejecting unique $r_1 \cdots r_n$ and accepting $x_c$, with appropriate renormalization at each step as we remove rejected samples.

- Our meta-proposal runs a second adaptive rejection loop, using the rejected samples from the first loop to further constrain the sampling space. Let $s = ((s_1, \dots, s_{n_1}), s^*) \in \mathcal{S}$, where $s_i$ are rejected samples, $s^*$ is the accepted sample, and $r \!+\! s$ denotes the concatenation of the rejected samples from both loops:

$$h(s, r; x)$$
$$= \frac{1}{n_0 + n_1 + 1} \prod_{i=1}^{n_0} (p_0(r_i)(1 - \mathbb{1}_{\mathcal{C}}(r_i))) \prod_{i=1}^{n_1} (p_0(s_i)(1 - \mathbb{1}_{\mathcal{C}}(s_i)))$$
$$\cdot p_0(s^*) \, \mathbb{1}_{\mathcal{C}}(s^*) \prod_i \frac{1}{\sum_{j=1}^{i} p_0(r \!+\! s)} \, \mathbb{1}_{[\forall i \neq j. (r \!+\! s)_i \neq (r \!+\! s)_j]} \qquad [Def.] \tag{29a}$$

$$= \frac{1}{n_0 + n_1 + 1} \prod_{i=1}^{n_0} p_0(r_i) \prod_{i=1}^{n_1} (p_0(s_i)(1 - \mathbb{1}_{\mathcal{C}}(s_i)))$$
$$\cdot p_0(s^*) \, \mathbb{1}_{\mathcal{C}}(s^*) \prod_i \frac{1}{\sum_{j=1}^{i} p_0(r \!+\! s)} \, \mathbb{1}_{[\forall i \neq j. (r \!+\! s)_i \neq (r \!+\! s)_j]} \qquad [\forall r \in \mathcal{R} : \mathbb{1}_{\mathcal{C}}(r) = 0] \tag{29b}$$

$$= \frac{1}{n_0 + n_1 + 1} \prod_{i=1}^{n_0} p_0(r_i) \prod_{i=1}^{n_1} p_0(s_i)$$

$$\cdot p_0(s^*) \, \mathbb{1}_{\mathcal{C}}(s^*) \prod_i \frac{1}{\sum_{j=1}^i p_0(r \pm s)} \, \mathbb{1}_{[\forall i \neq j.(r \pm s)_i \neq (r \pm s)_j]} \quad [\forall s^{(i)} \in \mathcal{S}: \mathbb{1}_{\mathcal{C}}\left(s^{(i)}\right) = 0] \quad (29\text{c})$$

$$= \frac{1}{n_0 + n_1 + 1} \prod_{i=1}^{n_0} p_0(r_i) \prod_{i=1}^{n_1} p_0(s_i)$$

$$\cdot p_0(s^*) \prod_i \frac{1}{\sum_{j=1}^i p_0(r \pm s)} \, \mathbb{1}_{[\forall i \neq j.(r \pm s)_i \neq (r \pm s)_j]} \quad [\forall s^* \in \mathcal{S}: \mathbb{1}_{\mathcal{C}}(s^*) = 1] \quad (29\text{d})$$

- Our meta-meta-proposal continues the adaptive rejection process, starting with the previously rejected samples in $r$ already removed from consideration:

$$j(s; r, x) = \left( \prod_{i=n_0+1}^{n_0+n_1} \frac{1}{1 - \sum_{j=1}^i p_0(r \pm s)} \right) \prod_{i=1}^{n_1} p_0(s_i)(1 - \mathbb{1}_{\mathcal{C}}(s_i))$$

$$\cdot p_0(s^*) \, \mathbb{1}_{\mathcal{C}}(s^*) \, \mathbb{1}_{[\forall i \neq j. s_i \neq s_j \wedge s_i \notin r]} \quad [\text{Def.}] \quad (30\text{a})$$

$$= \left( \prod_{i=n_0+1}^{n_0+n_1} \frac{1}{1 - \sum_{j=1}^i p_0(r \pm s)} \right) \prod_{i=1}^{n_1} p_0(s_i)$$

$$\cdot p_0(s^*) \, \mathbb{1}_{\mathcal{C}}(s^*) \, \mathbb{1}_{[\forall i \neq j. s_i \neq s_j \wedge s_i \notin r]} \quad [\forall s^{(i)} \in \mathcal{S}: \mathbb{1}_{\mathcal{C}}\left(s^{(i)}\right) = 0] \quad (30\text{b})$$

$$= \left( \prod_{i=n_0+1}^{n_0+n_1} \frac{1}{1 - \sum_{j=1}^i p_0(r \pm s)} \right) \prod_{i=1}^{n_1} p_0(s_i)$$

$$\cdot p_0(s^*) \, \mathbb{1}_{[\forall i \neq j. s_i \neq s_j \wedge s_i \notin r]} \quad [\forall s^* \in \mathcal{S}: \mathbb{1}_{\mathcal{C}}(s^*) = 1] \quad (30\text{c})$$

Following Def. 5, and substituting our definitions, we calculate the proper weight. Note that most terms of $p_0$ and $\mathbb{1}_{\mathcal{C}}$ will cancel out, as will most of the renormalization terms. The key difference from WRS is that when $q$ proposes the *accepted* value $x$ from the distribution $\frac{p_0(x)}{1 - \sum_{j=1}^{n_0} p_0(r)}$, there is no corresponding $\frac{1}{1 - \sum_{j=1}^{n_0} p_0(r)}$ term in the numerator:

$$w = \frac{\widetilde{p}(x) h(s, r; x)}{q(r, x) j(s; r, x)} \quad [\text{Def. 5}] \quad (31\text{a})$$

$$= \frac{p_0(x) \mathbb{1}_{\mathcal{C}}(x) \frac{1}{n_0+n_1+1} \cdot (\cdots)}{\left( \prod_{i=1}^{n_0} \frac{1}{1 - \sum_{j=1}^{i-1} p_0(r)} \right) \frac{p_0(x)}{1 - \sum_{j=1}^{n_0} p_0(r)} \cdot (\cdots)} \quad [\text{Substitution}] \quad (31\text{b})$$

$$= \frac{\frac{1}{n_0+n_1+1}}{\frac{1}{1 - \sum_{j=1}^{n_0} p_0(r)}} \quad [\text{Cancellation; AC}] \quad (31\text{c})$$

$$= \frac{1 - \sum_{j=1}^{n_0} p_0(r)}{n_0 + n_1 + 1} \quad [\text{Algebra}] \quad (31\text{d})$$

$$= \frac{1 - \psi_0}{n_0 + n_1 + 1} \quad [\psi_0 = \sum_{j=1}^{n_0} p_0(r)] \quad (31\text{e})$$

$$= \frac{1 - \psi_0}{n + 1} \quad [\text{Def. of } n] \quad (31\text{f})$$

This matches the weight formula in Def. 2, confirming that our adaptive sampling procedure is properly weighted for $\widetilde{p}$ as justified by the RAVI framework. ∎

# E  NumPy Implementations of Samplers (Defs. 1 and 2)

Note that implementations in this section are designed for helping readers develop intuition, tightly coupling math and code. This is opposed to efficient implementation and numerical stability, so such examples should be thoroughly adapted for performant use.

---

**Listing 1** Weighted Rejection Sampling (WRS)

```
6  def wrs(p₀, 𝟙_𝒞, L):
7      for i in range(L+1):
8          n_i = 0
9          while True:
10             x ~ p₀
11             if 𝟙_𝒞(x):
12                 if i == 0:
13                     x_c = x
14                 break
15             n_i += 1
16     Ẑ = L / (L + Σ_{i=0}^{L} n_i)
17     return ⟨x_c, Ẑ⟩
```

```
18 def wrs(p, cond, L):
19     n = np.zeros(L+1)
20     for i in range(L+1):
21         while True:
22             x = sample(p)
23             if cond(x):
24                 if i == 0:
25                     xc = x
26                 break
27             n[i] += 1
28     Zhat = L / (L + n.sum())
29     return xc, Zhat
```

---

**Listing 2** Adaptive Weighted Rejection Sampling (AWRS)

```
30 def awrs(p₀, 𝟙_𝒞):  # L = 1
31     n, ψ₀, r = 0, 0, set()
32     for i in range(2):
33         while True:
34             x ~ p₀(· | x ∉ r)
35             if 𝟙_𝒞(x):
36                 if i == 0:
37                     x_c = x
38                 break
39             else:
40                 if i == 0:
41                     ψ₀ += p₀(x)
42                 r.add(x)
43             n += 1
44     Ẑ = (1 − ψ₀) / (n + 1)
45     return ⟨x_c, Ẑ⟩
```

```
46 def awrs(p, cond):  # L = 1
47     n, psi0, r = 0, 0, set()
48     for i in range(2):
49         while True:
50             x = cond_sample(p, r)
51             if cond(x):
52                 if i == 0:
53                     xc = x
54                 break
55             else:
56                 if i == 0:
57                     psi0 += p[x]
58                 r.add(x)
59         n += 1
60     Zhat = (1 - psi0) / (n + 1)
61     return xc, Zhat
```

# F  Simulation Results of Samplers (Defs. 1 and 2)

In this section, we report empirical results from simulation that illustrate the unbiasedness as well as favorable variance and runtime of our algorithms.

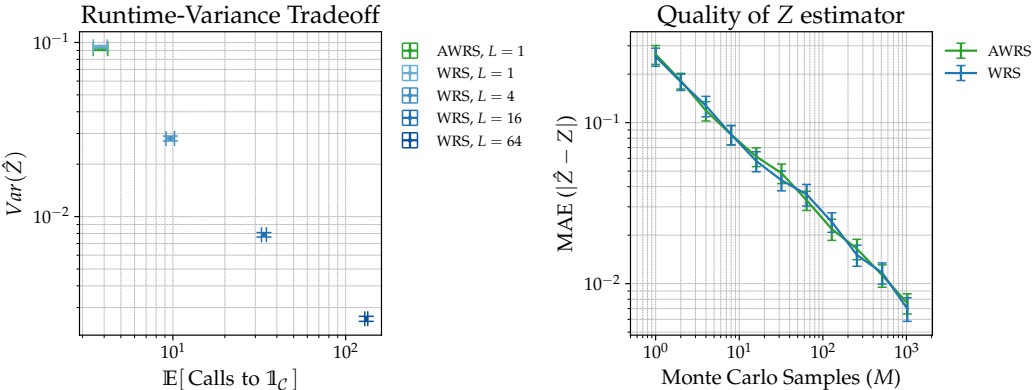

Figure F.1: Left: The compute budget of Def. 1 (WRS) can trade off runtime and variance. Def. 2 (AWRS) shows comparable variance to WRS at $L = 1$, which we find sufficient in practice. Right: The algorithms described in Def. 1 (WRS; $L$=1) and Def. 2 (AWRS) are properly weighted. As we draw increasingly many Monte Carlo samples ($M$) from each sampler, the mean absolute error (MAE) in the estimate of $Z$ trends towards 0. Across both panels, error bars reflect 95% confidence intervals over an outer loop of 100 Monte Carlo iterations, where $p_0 \sim D_V(\cdot \mid \mathbf{1})$, a uniform Dirichlet draw of probability vectors with size $V = 1,000$, and $\mathbb{1}_C \sim B_V(\cdot; \text{Uniform}(0,1))$, a draw from a Bernoulli process of size $V$ with success probability $\pi \sim \text{Uniform}(0,1)$.

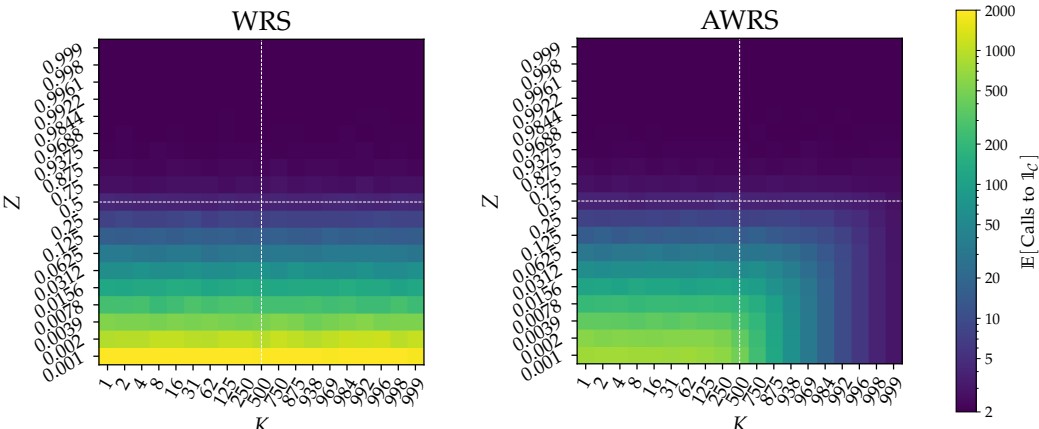

Figure F.2: AWRS runtime scales favorably. Here we visualize Monte Carlo ($N = 100$ per cell) estimates of expected runtime and "corner-case" behavior. As $Z$ increases, runtime decreases. As $K$ (the number of tokens conforming to $\mathbb{1}_C$) increases towards $V = 1,000$ (the vocabulary size), runtime decreases for AWRS. Note that the scales of $Z$ and $K$ decay logarithmically towards the corners and the color axis also scales logarithmically. For AWRS, in the regime where $K$ is small, runtime operates dominantly on $Z$. As $K$ gets large, runtime is capped by $V - K$.

# G Runtime Analysis

## G.1 Runtime Analysis of Def. 1 (WRS)

**Proposition 2.** *The expected runtime of $Q_{WRS}$ scales with $\mathcal{O}(\frac{L}{Z})$.* *(App. G.1)*

*Proof.* Def. 1 performs $L + 1$ independent rejection sampling loops. In each loop, we sample $x_j \sim p_0$ repeatedly, until obtaining a sample such that $\mathbb{1}_{\mathcal{C}}(x_j) = 1$.

Define total runtime as $T \stackrel{\text{def}}{=} \sum_{i=0}^{L} T_i$, where random variable $T_i$ denotes the number of samples in the $i$th loop (some number $n_i \geq 0$ of rejected samples, and one final successful sample).

Since we sample independently with replacement, each $T_i$ is identically distributed as a geometric distribution supported on $\mathbb{N}_{\geq 1}$ with parameter $Z = \Pr_{x \sim p_0}[\mathbb{1}_{\mathcal{C}}(x)] \in (0, 1]$. Thus total runtime is as follows.

$$\mathbb{E}[T] = \sum_{i=0}^{L} \mathbb{E}[T_i] \qquad \textit{[Def. 1; Linearity of expectation]} \tag{32a}$$

$$= \sum_{i=0}^{L} \sum_{k=1}^{\infty} k \, \Pr[T_i = k] \qquad \textit{[Def. of expectation]} \tag{32b}$$

$$= (L+1) \sum_{k=1}^{\infty} k \, (1-Z)^{k-1} Z \qquad \textit{[PMF of geom. dist.; Independence]} \tag{32c}$$

$$= (L+1) \sum_{k=1}^{\infty} k r^{k-1} (1-r) \qquad \textit{[Let } r = 1 - Z\textit{]} \tag{32d}$$

$$= (L+1) \sum_{k=1}^{\infty} r^{k-1} \qquad \textit{[Simplify by subtracting like terms]} \tag{32e}$$

$$= (L+1) \frac{1}{1 - (1-Z)} \qquad \textit{[Sum geom. series; } r = 1 - Z < 1\textit{]} \tag{32f}$$

$$= \frac{L+1}{Z} \tag{32g}$$

Thus, the computational complexity is $\mathcal{O}\left(\frac{L}{Z}\right)$. ∎

## G.2 Runtime Analysis of Def. 2 (AWRS)

**Proposition 4.** *The expected runtime of $Q_{AWRS}$ scales with $\mathcal{O}(\sum_{x \notin \mathcal{C}} \pi_x)$, where $\pi_x \stackrel{\text{def}}{=} \frac{p_0(x)}{p_0(x) + Z}$, i.e., the probability of each non-conforming token relative to Z.* *(App. G.2)*

*Proof.* In AWRS, we sample *without replacement* from initial distribution $p_0$, until we reach a sample which satisfies the constraint, which is sampled *with replacement*. This is repeated for a total of $1 + L$ successive loops.

To derive a simple expression for the expected total number of samples, Eq. (42), the following general property will be useful.

**Lemma 1.** *Let $\widetilde{P}$ be any (potentially unnormalized) probability distribution over the discrete set $\mathcal{X}$. Suppose we begin sequentially sampling (with or without replacement) according to $\widetilde{P}$. For any two disjoint sets $\mathcal{A}_1, \mathcal{A}_2 \subseteq \mathcal{X}$, let $x_{\mathcal{A}}$ be the first sampled element that happens to be in $\mathcal{A} \stackrel{\text{def}}{=} \mathcal{A}_1 \sqcup \mathcal{A}_2$. The probability that x is in $\mathcal{A}_1$ (write $\mathcal{A}_1 \prec \mathcal{A}_2$) is $\Pr[\mathcal{A}_1 \prec \mathcal{A}_2] = \frac{\widetilde{P}(\mathcal{A}_1)}{\widetilde{P}(\mathcal{A})}$.*

*Proof.* The crucial insight is that for a sequence drawn with or without replacement, the event $\mathcal{A}_1 \prec \mathcal{A}_2$ is determined by the first occurrence of an element in $\mathcal{A}$ (denote this element $x_{\mathcal{A}}$), which is independent of the order or weight of elements outside of $\mathcal{A}$. That is, $\Pr[\mathcal{A}_1 \prec \mathcal{A}_2] = \Pr[x_{\mathcal{A}} \in \mathcal{A}_1] = \frac{P'(\mathcal{A}_1)}{P'(\mathcal{A})}$, where $P'$ is the probability distribution from which $x_{\mathcal{A}}$ is sampled.

In sampling with replacement the distribution $P' = P \propto \widetilde{P}$ trivially, and we have our result. In sampling without replacement, let $R \subset \mathcal{X}$ be the set of already-rejected elements at the point when $x_{\mathcal{A}}$ is drawn. Then the sampling distribution is $P'(x) \propto \mathbb{1}_{\mathcal{X}\setminus R}(x) P(x)$. Since the elements in $\mathcal{A}$ are not yet rejected by definition, simply $\frac{P'(\mathcal{A}_1)}{P'(\mathcal{A})} = \frac{P(\mathcal{A}_1)}{P(\mathcal{A})} = \frac{\widetilde{P}(\mathcal{A}_1)}{\widetilde{P}(\mathcal{A})}$. ∎

Define total runtime as the total number of samples (both rejected and accepted) in the algorithm:

$$T \stackrel{\text{def}}{=} T_0 + T_1 + \cdots + T_L \tag{33}$$

where $T_0$ is the runtime of the first loop (number of samples up to and including the first accepted sample), and $T_i$ is the runtime of the $i^{\text{th}}$ additional loop, for $i \in \{1, \ldots, L\}$. $T$ is the number of calls to $\mathbb{1}_{\mathcal{C}}$ in the algorithm.

Applying Lemma 1 in our setting, within the first loop, where we are sampling without replacement from initial distribution $p_0$, define the following shorthand for the probability that $x \in \mathcal{X} \setminus \mathcal{C}$ appears in the sequence of samples before any element from the set $\mathcal{C}$.

$$\pi_x \stackrel{\text{def}}{=} \Pr[x \prec \mathcal{C}] = \frac{p_0(x)}{p_0(x) + \sum_{x' \in \mathcal{C}} p_0(x')} = \frac{p_0(x)}{p_0(x) + Z} \tag{34}$$

Note that each $\pi_x$ is strictly less than 1, since by assumption $p_0$ places some nonzero probability mass on elements in $\mathcal{C}$.

**Runtime of First Loop:** $T_0$   For each $x \in \mathcal{X} \setminus \mathcal{C}$ define an indicator random variable $Y_x^{(0)} = 1$ if $x$ appears in the loop, else 0. The runtime $T_0$ is the number of rejections, plus one accepted trial:

$$T_0 \stackrel{\text{def}}{=} 1 + \sum_{x \notin \mathcal{C}} Y_x^{(0)} \tag{35}$$

Thus the expected runtime is as follows.

$$\mathbb{E}[T_0] = 1 + \sum_{x \notin \mathcal{C}} \mathbb{E}[Y_x^{(0)}] \qquad \text{[Linearity of expectation]} \tag{36a}$$

$$= 1 + \sum_{x \notin \mathcal{C}} \Pr[Y_x^{(0)} = 1] \qquad \text{[Expectation of indicator RV]} \tag{36b}$$

$$= 1 + \sum_{x \notin \mathcal{C}} \pi_x \qquad \text{[Definition of } Y_x\text{'s and Lemma 1]} \tag{36c}$$

**Runtime of a Subsequent Loop:** $T_i$   Now consider the $i^{\text{th}}$ loop, for $i \geq 1$.

Denote $R_i \subseteq (\mathcal{X} \setminus \mathcal{C})$ the set of accumulated rejections from previous loops, and $\psi_i \stackrel{\text{def}}{=} \sum_{r \in R_i} p_0(r)$ the total already-rejected probability mass. Note $\psi_i < 1$ by assumption. We begin the $i^{\text{th}}$ loop sampling from a distribution which assigns probability $\frac{p_0(x)}{1-\psi_i}$ to each $x \in \mathcal{X} \setminus R_i$.

Given $R_i$, for each $x \in \mathcal{X} \setminus (\mathcal{C} \cup R_i)$ define indicator random variable $Y_x^{(i)} = 1$ if $x$ appears in the current loop, else 0. Then,

$$\mathbb{E}[Y_x^{(i)} \mid R_i] = \Pr[Y_x^{(i)} = 1 \mid R_i] \qquad \textit{[Expectation of indicator RV]} \qquad (37a)$$

$$= \frac{\frac{p_0(x)}{1-\psi_i}}{\frac{p_0(x)}{1-\psi_i} + \frac{Z}{1-\psi_i}} \qquad \textit{[Definition of $Y_x^{(i)}$ and Lemma 1]} \qquad (37b)$$

$$= \frac{p_0(x)}{p_0(x) + Z} \qquad \textit{[Cancellation of $1 - \psi_i > 0$]} \qquad (37c)$$

$$= \pi_x \qquad \textit{[Definition of $\pi_x$]} \qquad (37d)$$

Importantly, the probability $\Pr[Y_x^{(i)} = 1 \mid R_i]$ is in fact independent of $R_i$, and is simply equal to $\pi_x = \Pr[Y_x^{(0)} = 1]$, for each $x \in \mathcal{X} \setminus (\mathcal{C} \cup R_i)$. That is, the probability is independent of the previous rejections, and independent of which loop we are in, provided only that $x$ is not in $\mathcal{C}$, and has not yet been rejected. Note this is equivalent to the probability that $x$ is rejected in the current loop, given it has not been rejected earlier:

$$\Pr[x \in R_{i+1} \mid x \notin (\mathcal{C} \cup R_i)] = \mathbb{E}[Y_x^{(i)} \mid R_i] = \pi_x \qquad (38)$$

Thus for any $x \in \mathcal{X} \setminus \mathcal{C}$, the probability that it has not yet been rejected is

$$\Pr[x \notin R_i \mid x \notin \mathcal{C}] = \prod_{i'=1}^{i} \Pr[x \notin R_{i'} \mid x \notin (\mathcal{C} \cup R_{i'-1})] = (1 - \pi_x)^i \qquad (39)$$

letting $R_0 \stackrel{\text{def}}{=} \varnothing$ for convenience, and noting $\Pr[x \notin R_1 \mid x \notin \mathcal{C}] = 1 - \mathbb{E}[Y_x^{(0)}] = 1 - \pi_x$.

Given already-rejected $R_i$ from previous runs, the runtime $T_i$ is the number of rejections, plus one accepted trial:

$$T_i \stackrel{\text{def}}{=} 1 + \sum_{x \notin (\mathcal{C} \cup R_i)} Y_x^{(i)} \qquad (40)$$

Using the properties above, we can derive the expected runtime in loop $i$ as follows.

$$\mathbb{E}[T_i] = \mathbb{E}\left[\mathbb{E}\left[T_i \mid R_i\right]\right] \qquad \textit{[Law of total expectation]} \qquad (41a)$$

$$= \mathbb{E}\left[1 + \sum_{x \notin (\mathcal{C} \cup R_i)} \mathbb{E}\left[Y_x^{(i)} \mid R_i\right]\right] \qquad \textit{[Linearity of expectation, defn of $T_i$]} \qquad (41b)$$

$$= \mathbb{E}\left[1 + \sum_{x \notin (\mathcal{C} \cup R_i)} \pi_x\right] \qquad \textit{[Eq. (37)]} \qquad (41c)$$

$$= \mathbb{E}\left[1 + \sum_{x \notin \mathcal{C}} \mathbb{1}_{\mathcal{X} \setminus R_i}(x) \pi_x\right] \qquad \textit{[Reindexing]} \qquad (41d)$$

$$= 1 + \sum_{x \notin \mathcal{C}} \mathbb{E}\left[\mathbb{1}_{\mathcal{X} \setminus R_i}(x)\right] \pi_x \qquad \textit{[Linearity of expectation]} \qquad (41e)$$

$$= 1 + \sum_{x \notin \mathcal{C}} \Pr[x \notin R_i] \pi_x \qquad \textit{[Expectation of indicator RV]} \qquad (41f)$$

$$= 1 + \sum_{x \notin \mathcal{C}} (1 - \pi_x)^i \pi_x \qquad \textit{[Eq. (39)]} \qquad (41g)$$

We can see from the derivation of $\mathbb{E}[T_0]$ in Eq. (36) that this expression also holds for $i = 0$.

**Total Expected Runtime**   Putting this together, we have that for $L \geq 1$, the expected total runtime is

$$\mathbb{E}[T] = \sum_{i=0}^{L} \mathbb{E}[T_i] \qquad\qquad \textit{[Linearity of expectation]} \quad (42a)$$

$$= \sum_{i=0}^{L} \left[ 1 + \sum_{x \notin \mathcal{C}} (1 - \pi_x)^i \pi_x \right] \qquad\qquad \textit{[Eq. (41)]} \quad (42b)$$

$$= 1 + L + \sum_{x \notin \mathcal{C}} \sum_{i=0}^{L} (1 - \pi_x)^i \pi_x \qquad \textit{[Sum constant; Change summation order]} \quad (42c)$$

$$= 1 + L + \sum_{x \notin \mathcal{C}} \frac{1 - (1 - \pi_x)^{L+1}}{1 - (1 - \pi_x)} \pi_x \qquad\qquad \textit{[Partial sum of geom. series]} \quad (42d)$$

$$= 1 + L + \sum_{x \notin \mathcal{C}} 1 - (1 - \pi_x)^{L+1} \qquad\qquad \textit{[}\pi_x > 0\textit{]} \quad (42e)$$

$$= 1 + L + |\mathcal{X} \setminus \mathcal{C}| - \sum_{x \notin \mathcal{C}} (1 - \pi_x)^{L+1} \qquad\qquad (42f)$$

From this we can see that our runtime decays polynomially in $L$ with respect to the $\pi_x$'s (where, recall, $\pi_x \overset{\text{def}}{=} \frac{p_0(x)}{p_0(x)+Z} < 1$), and so each additional loop comes at reduced cost.[16] Note that once all items that do not satisfy the constraint are removed, there is just a single step per sampling loop. This corresponds to the intuition that as we add subsequent loops of sampling without replacement, the expected number of samples before success in each loop decreases as the probability mass is shifted off of already-ruled-out items, and the runtime approaches linear growth in $L$.

More formally, that is: Considering the behavior as $L$ grows, $(1 - \pi_x)^{L+1}$ approaches 0, so the runtime scales as $\mathbb{E}[T] = \mathcal{O}(L)$.

For fixed $L$, the runtime scales in the probabilities $\pi_x$, as $\mathbb{E}[T] = \mathcal{O}(\sum_{x \notin \mathcal{C}} \pi_x)$, since higher powers are dominated by the linear term for each $\pi_x$. [Also, considering the behavior as $\pi_x \to 1$ for each $x \in \mathcal{X} \setminus \mathcal{C}$ (that is, when the probability mass on items satisfying the constraint becomes negligible in pairwise comparison to those each of those that don't), we again have that $(1 - \pi_x)^{L+1} \to 0$. So the expected runtime approaches $1 + L + |\mathcal{X} \setminus \mathcal{C}|$.]

Note, for the special case $L = 1$, the expected runtime is

$$\mathbb{E}[T] = \mathbb{E}[T_0] + \mathbb{E}[T_1] \qquad\qquad (43a)$$

$$= \left[ 1 + \sum_{x \notin \mathcal{C}} \pi_x \right] + \left[ 1 + \sum_{x \notin \mathcal{C}} \pi_x (1 - \pi_x) \right] \qquad\qquad (43b)$$

$$= 2 + \sum_{x \notin \mathcal{C}} 2\pi_x - \pi_x^2 \qquad\qquad (43c)$$

∎

---

[16]Empirically, in the context-sensitive pattern matching domain, the second loop samples only 1 token in 92%, 85%, and 72% of runs for Llama 70B, 8B, and 1B, respectively.

## G.3 Runtime Simulation Comparison

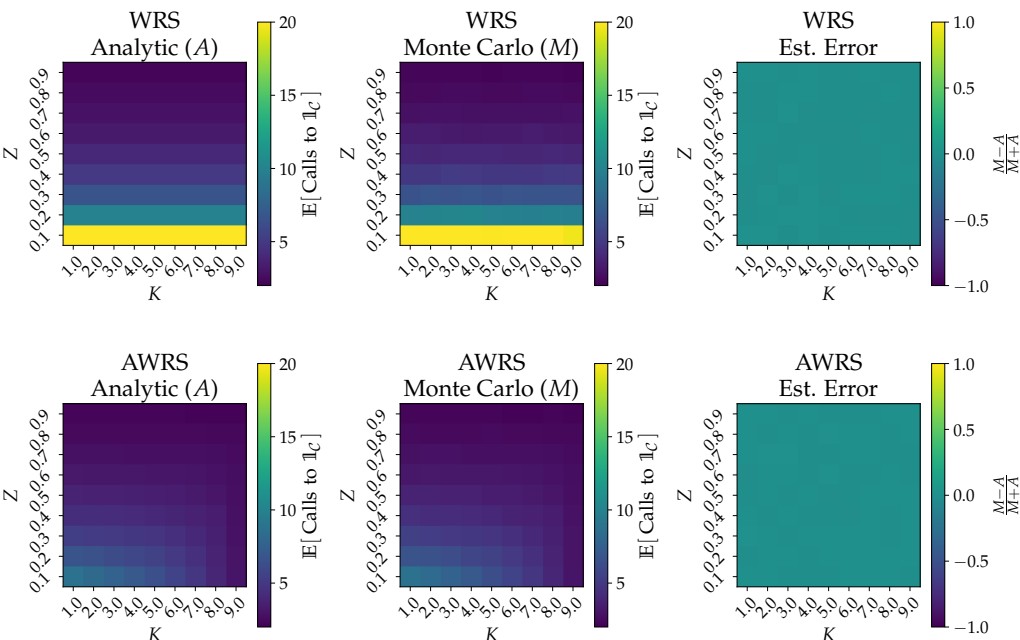

Figure G.1: Analytic solutions to expected runtime are corroborated by empirical simulation. Here we visualize Monte Carlo ($N = 1,000$ per cell) estimates of expected runtime as a function of $Z$ and $K$ (the number of tokens conforming to $\mathbb{1}_{\mathcal{C}}$) for a dense tiling of a vocabulary of size $V = 10$. Note again that AWRS scales more gracefully than WRS.

# H  Extensions of Def. 2 (AWRS)

## H.1  Partial Concurrency

The algorithm described in Def. 2 can be further sped up by executing the rejection loop concurrently. The key insight is that since we never resample any rejected element, as long as we encounter rejections, we may concurrently sample-without-replacement (SWOR), only syncing between loops. The recipe for partially concurrent AWRS is as follows:

**Definition 6.** *Given an unnormalized target $\widetilde{p}$, concurrent AWRS generates $\langle x, \widehat{Z} \rangle \sim Q_{AWRS}$ as follows:*

1. *Sample a collection of keys $\kappa_x \sim \mathrm{Exp}(p_0(x))$, i.e., independent samples from exponential distributions whose rates are parameterized by $p_0$.*
2. *Sort the tokens by their keys, $x_1, \dots, x_{|\mathcal{X}|}$ such that $\kappa_{x_1} \leq \cdots \leq \kappa_{x_{|\mathcal{X}|}}$.*
3. *Evaluate $\mathbb{1}_{\mathcal{C}}(x_1), \dots, \mathbb{1}_{\mathcal{C}}(x_{|\mathcal{X}|})$ concurrently until the first acceptance at $x_i$.*
4. *Kill all threads with index $> i$ and wait for threads with index $< i$.*
5. *Set $x$ to the min index passing element $x_j$, set $n_0$ to $j - 1$, and set $\psi_0$ to $\sum_{i=0}^{j-1} p_0(x)$.*
6. *Mark elements $x_1, \dots, x_{j-1}$ before the next loop.*
7. *Repeat for the next loop, still counting rejections as in Def. 2.*
8. *Calculate $\widehat{Z}$ as usual.*

As an additional minor speed-up, since all exponential perturbations are independent, we may pre-sample them in (very large) batches and simply stream them as needed in the loop.

For unnormalized logits $\log \widetilde{p_0}$, sample keys $\kappa_x \sim \mathrm{Gumbel}(\log \widetilde{p_0})$ and sort keys descending rather than ascending (Vieira, 2014).

## H.2  Clipped-Probability Early Stopping

The algorithm described in Def. 2 may take up to $|\mathcal{X} \setminus \mathcal{C}|$ steps to find a token, and as more mass is removed, $\widehat{Z}$ may get arbitrarily small. In these cases, we risk spending compute on samples that will likely be dropped by SMC, anyways. Here, we present a strategy for early stopping in such cases, based on a pair of thresholds on rejected probability mass, one for each loop. While this speedup can be beneficial in some settings, clipped AWRS is no longer an exact sampler and may yield dead ($\widehat{Z} = 0$) samples.

**Definition 7.** *Given an unnormalized target $\widetilde{p}$, **clipped AWRS** generates $\langle x, \widehat{Z} \rangle \sim Q_{CAWRS}$ as follows:*

1. *Select two thresholds $0 < \theta_0 < \theta_1 < 1$.*
2. *Sample $\langle r_1, \dots, r_{n_0}, \dots \rangle$ as follows: draw $n_0$ unique rejections through a sequence of renormalized distributions on $\mathcal{X} \setminus \mathbf{r}_{<i}$ until either obtaining $x_{c_0} \in \mathcal{C}$ or $\sum_{i=1}^{n_0} p_0(r_i) > \theta_0$.*
3. *If $x_{c_0}$ was found, generate an additional trace $\langle s_1, \dots, s_{n_1}, \dots \rangle$, by continuing to sample as above from the remaining not-yet-rejected elements, through an additional $n_1$ new unique rejections until either a token is accepted or $\sum_{i=1}^{n_0} p_0(r_i) + \sum_{j=1}^{n_1} p_0(s_j) > \theta_1$. Then calculate the standard $\widehat{Z} = \frac{1-\psi_0}{n+1}$, where $\psi_0 = \sum_{i=1}^{n_0} p_0(r_i)$ and $n = n_0 + n_1$.*
4. *If no $x_{c_0}$ was found before hitting $\theta_0$, draw 1 single sample $x^*$ from the remaining elements after the first loop $\mathcal{X} \setminus \mathbf{r}_{<i}$.*
5. *If $\mathbb{1}_{\mathcal{C}}(x^*) = 0$, then $\widehat{Z} = 0$.*
6. *If $\mathbb{1}_{\mathcal{C}}(x^*) = 1$, then run the second loop as in step 3 through either an acceptance or the $\theta_1$ cutoff, but set $\widehat{Z} = \frac{(n_1+1)(1-\psi_0)}{n+1}$, where $n_1$ is the number of rejections only in the second loop.*

*In summary, $\widehat{Z} \stackrel{\mathrm{def}}{=} \frac{(n_1+1)^{\mathbb{1}[\psi_0 > \theta_0]}(1-\psi_0)}{n+1}$*

### H.2.1  RAVI Perspective

We may think of clipped AWRS as the following extension of AWRS (App. B.6):

**Proposal.** We again define over the space $\mathcal{X} \times \mathcal{R}$, and as before, run an adaptive rejection loop. However, we now stop *either* when we see an accepted sample $x_{c_0}$, *or* when the rejected samples have total probability mass exceeding $\theta_0$. In that case, the returned $x$ is simply the next proposed value, whether or not it satisfies the constraint.

**Meta-proposal.** Given $x$, we only care when $x$ satisfies the constraint; otherwise, no matter our meta-proposal, $\widehat{Z} = 0$. Assuming $x$ does satisfy the constraint, the meta-proposal generates its own rejection loop with probability mass bound by $\theta_1$, then selects a split point $n_0$. If the total mass up to the split point exceeds $\theta_0$, we clamp $n_0$ to the first index at which $\theta_0$ was exceeded. This allows us to tractably infer the first loop, once again not needing to actually sample the split point.

**Meta-meta-proposal.** We continue generating a second loop with threshold $\theta_1$ to propose the remaining samples from the meta-proposal, which we had stopped at its split point.

### H.2.2 Practical considerations

Stochastic runtimes (particularly if the runtime distribution has long tails) can lead to issues in production systems. Here, we have presented a modified version of AWRS that incorporates a user-specified upper bound on the mass sampled to generate a satisfying next token. Using this variant, if a single particle within SMC has found itself in a poor position, it will stop early with a weight of 0, allowing resampling to cull the particle (and clone a more promising particle) without slowing down the rest of the algorithm. If a constraint is so difficult that the upper bound is hit frequently (across all particles in an SMC algorithm), the user may opt to increase their upper bound, dispatch to a larger model more capable of satisfying the constraint, or increase the number of particles in SMC to mitigate the issue. We recommend testing how the inclusion of early stopping at various levels of strictness affect accuracy/runtime tradeoffs in your exact setting.

### H.3 Bounded cost early stopping

Another way to implement early stopping is to set an upper bound on the number of rejected tokens seen. In this section we present three different algorithms for accomplishing this. The first is a variant of the with-replacement WRS algorithm (Def. 1), which we present as a foundation. The second two are variants of AWRS (Def. 2), each with slightly different performance characteristics. Each takes a parameter $R$ and guarantees that no more than $R + L + 1$ calls to the constraint function will be made (up to $R$ of which will fail to satisfy the constraint and up to $L + 1$ satisfy the constraint).

### H.3.1 Clipped Weighted Rejection Sampling

**Definition 8.** *Given an unnormalized target $\widetilde{p}$, **clipped WRS** generates $\langle x, \widehat{Z} \rangle \sim Q_{CWRS}$ as follows:*

1. *Sample with replacement from $p_0$ to get a sequence $\langle x_1, \ldots, x_n \rangle$, stopping when you have observed either $L + 1$ accepted samples (not-necessarily-unique elements $x_i$ with $\mathbb{1}_{\mathcal{C}}(x_i) = 1$) or $R$ rejected samples (not-necessarily-unique elements $x_j$ with $\mathbb{1}_{\mathcal{C}}(x_j) = 0$).*
2. *Let $s$ be the number of accepted samples and $r$ be the number of rejected samples so $n = r + s$ is the total number of samples seen. If $s = L + 1$ (i.e. we stopped on an accepted element), let $\widehat{Z} = \frac{s-1}{n+1}$. Otherwise (i.e. $r = R$ and we stopped on a rejected element), let $\widehat{Z} = \frac{s}{n+1}$.*
3. *If $s > 0$ let $x$ be the first accepted token, otherwise let $x = x_1$ (the choice in the $s = 0$ case is arbitrary, but choosing a token we have already sampled ensures that we never return an accepted token with a 0 weight, which is a convenient property).*
4. *Return $\langle x, \widehat{Z} \rangle$*

The proof of the correctness of this estimator follows from applying the Rao-Blackwell theorem to the estimator $\mathbb{1}_{\mathcal{C}}(x_1)$ (i.e., whether the first sampled token satisfies the constraint) conditioned on the sufficient statistic $(r, s)$ and counting the number of sequences that can

have a given value $(r, s)$. This is essentially the same as the proof of the standard negative binomial result (see App. C.1).

### H.3.2 Geometric Adaptive Weighted Rejection Sampling

We can now reduce the sampling without replacement version to the above, through the following observation: Any run of sampling without replacement can be thought of as sampling with replacement, where any samples that have already been seen during the run are discarded (not recorded). Thus you can construct a run of sampling with replacement from a run of sampling without replacement by inserting a number of "phantom" samples that the without replacement algorithm removed. Because the estimator only depends on the *number* of tokens, we do not need to actually generate these phantom tokens, only to count them, so instead of actually sampling individual phantom tokens (which is slow), you can sample from a geometric distribution representing the number of phantoms to insert before getting a given novel token.

**Definition 9.** *Given an unnormalized target $\widetilde{p}$, **geometric adaptive weighted rejection sampling** (GAWRS) generates $\langle x, \widehat{Z} \rangle \sim Q_{GAWRS}$ as follows:*

1. *Sample a sequence $\langle x_1, \dots, x_i, \dots \rangle$ from $p_0$ without replacement of rejected tokens (i.e., samples are drawn proportional to the probabilities of not-yet-rejected tokens).*
2. *Maintain a sequence $q_1, \dots, q_i, \dots$ representing the total probability mass removed prior to sampling $x_i$ (so $q_1 = 0$).*
3. *Maintain counters $s_0, s_1, \dots, s_i, \dots$ and $r_0, r_1, \dots, r_i, \dots$ representing the total number of accepted and rejected tokens seen so far, with $s_i$ defined as simple counting (i.e. $s_0 = 0, s_i = \mathbb{1}_{\mathcal{C}}(x_i) + s_{i-1}$), but $r_i$ defined to also count the phantom tokens seen so far, so $r_0 = 0$, but $r_i \sim r_{i-1} + \text{Geom}(q_i) + 1 - \mathbb{1}_{\mathcal{C}}(x_i)$.*
4. *Stop when $r_i \geq R$ or $s_i = L + 1$. Let $s = s_i$, $r = \min(R, r_i)$ (because if $r_i > R$ we should have stopped in the middle of a run of phantom tokens), let $n = r + s$ and use the same estimator as in Def. 8 above: If $s = L + 1$, let $\widehat{Z} = \frac{s-1}{n+1}$. Otherwise, let $\widehat{Z} = \frac{s}{n+1}$.*
5. *If $s > 0$ let $x$ be the first accepted token, otherwise let $x = x_1$.*
6. *Return $\langle x, \widehat{Z} \rangle$*

This method, without the early stopping, can also be used to provide an alternative form of the main AWRS algorithm. That being said, preliminary experimentation suggests that this approach is slightly more expensive and does not yield meaningfully different variance.

### H.3.3 Recursive Adaptive Weighted Rejection Sampling

Another method operates from the trivial observation that $Z = \mathbb{E}[\mathbb{1}_{\mathcal{C}}(X)] = p_0(x)\mathbb{1}_{\mathcal{C}}(x) + (1 - p_0(x))\mathbb{E}[\mathbb{1}_{\mathcal{C}}(X)|X \neq x]$, for $X \sim p_0$ and arbitrary $x \in \mathcal{X}$. This allows us to recursively construct an estimator for $Z$ by picking some $x$, then applying either some "base case" estimator or recursively applying the same formula for $\mathbb{E}[\mathbb{1}_{\mathcal{C}}(X)|X \neq x]$.

In particular, if we take our sampling without replacement loop as providing our choices of $x$, the remaining values are precisely sampled from the conditional distribution.

**Definition 10.** *Given an unnormalized target $\widetilde{p}$, **recursive adaptive weighted rejection sampling** (RAWRS) generates $\langle x, \widehat{Z} \rangle \sim Q_{RAWRS}$ as follows:*

1. *Sample without replacement from $p_0$ to get a sequence $\langle x_1, \dots, x_n \rangle$ stopping when you find the first $x_n$ with $n \leq R$ and $\mathbb{1}_{\mathcal{C}}(x_n) = 1$.*
2. *For each $x_i$, let $q_i$ be its conditional probability: $q_i \stackrel{\text{def}}{=} \frac{p_0(x_i)}{1 - \sum\limits_{j<i} p_0(x_i)}$. Let $m_i = \prod\limits_{j<i}(1 - q_i)$.*
3. *If $n = R$ then if $\mathbb{1}_{\mathcal{C}}(x_n) = 1$ let $\widehat{Z} = m_i$, otherwise let $\widehat{Z} = 0$.*
4. *If $n < R$ then if $\mathbb{1}_{\mathcal{C}}(x_{n+1}) = 1$ let $\widehat{Z} = m_i$, otherwise let $\widehat{Z} = q_i m_i$.*
5. *Let $x = x_n$.*
6. *Return $\langle x, \widehat{Z} \rangle$*

This works by recursively applying the formula until we've drawn $R - 1$ samples that don't satisfy the constraint or 1 that has, then using as the base case the trivial estimator that samples a single token and returns 1 if it satisfies the constraint and 0 if it doesn't.

### H.3.4 Trade-offs and Performance Considerations

Which of these two algorithms performs better depends on the shape of the distribution. In terms of absolute number of calls, RAWRS always makes the same or strictly fewer calls than GAWRS, but this ignores the fact that some of the calls in the latter may be repeated and not need recalculating (if the same accepted token is sampled multiple times). In particular, in the case where the distribution has a single element $x$ with $\mathbb{1}_\mathcal{C}(x) = 1$ and $p_0(x) > \frac{2}{3}$, the expected number of calls made by GAWRS is under 2, while the minimum number of calls made by RAWRS is always at least 2.

This $\frac{2}{3}$ threshold follows from the fact that the number of calls to the constraint function is bounded above by one plus the number of tokens other than $x$ seen before $x$ is drawn twice (one call for evaluating $\mathbb{1}_\mathcal{C}(x)$ and at most one call per other token seen), which is bounded above by a negative binomial distribution with parameters $(2, p_0(x))$ (because that's how many you'd see in the with-replacement model). So the expected number of calls is at most $1 + \frac{2(1 - p_0(x))}{p_0(x)}$, which some simple algebra shows is strictly less than two when $p_0(x) > \frac{2}{3}$.

In contrast, in less peaked distributions $p_0$, GAWRS may often need to sample more than 1 token in its second loop, and thus RAWRS will strictly dominate in those cases.

These two methods can also be hybridized: At any point in RAWRS one can switch to GAWRS in the recursion. This likely yields the best of both worlds, but considerable further investigation is needed to empirically explore this design space, which we leave for future work.

### H.4 Combination with truncation sampling

There are a few ways one might combine AWRS with truncation-based modifications to an LM's sampling procedure.

The most straightforward way is to think of truncation-based sampling as modifying the base LM distribution, on top of which AWRS can then be applied. For example, in top-p sampling with $p = 0.9$, we zero out the bottom 10% of token probabilities and renormalize to obtain a modified next-token probability distribution. We can then apply AWRS as though the base LM had generated this truncated next-token probability distribution: we adaptively rejection-sample from the truncated distribution, rather than from the full next-token distribution. One caveat is that the user must ensure that truncation does not remove all valid next tokens (according to the constraint) from the support.

Another way to combine AWRS with truncation-based sampling is to consider the truncation as part of the constraint: we write a new constraint function that accepts a token only if it passes the original constraint and is in the top 10% of logits from the base model. If AWRS-SMC were applied with this constraint, the weights would correct not only for myopia from locally constrained decoding, but also for myopia from top-p decoding. As the number of particles grows, the algorithm's output distribution would converge to the global posterior distribution that arises when the base LM is conditioned on the event that (1) the string satisfies the original constraint, and (2) each token is in the top 10% of the LM's predictions given the previous tokens. Note that, as above, the user must be careful to avoid the situation where there are no strings that satisfy this more stringent constraint.

# I Constraint Details

## I.1 Constraint Costs

| Domain | Cost (ms/eval) |
|---|---|
| Goal inference | 11.0 |
| Molecular synthesis | 0.30 |
| Pattern matching | 0.0054 |
| Text-to-SQL | 6.5 |
| JSON | 0.47 |

Table 2: The evaluation cost of each domain's local constraint checker.

The cost of constraint evaluation varies dramatically depending on constraint complexity. Tab. 2 presents the mean cost (ms/eval) of the constraints tested in this paper. As can be seen, costs range from $< 0.1$ms to $> 10$ms, and yet all can still be integrated into the decoding process with ARS-LCD, incurring a comparatively low constant-factor overhead (mean $1.22\times$ sec/ex across our benchmarks). Contrast this with TM-LCD, which evaluates the constraint on all tokens at each step. This method is so costly that we could only tractably run the full token masking baseline for the context-sensitive pattern matching domain. Even there, the approach incurs $> 50\times$ overhead.

Alternatively, most existing implementations of token masking, including libraries such as Outlines (Willard & Louf, 2023) and Guidance (Lundberg et al., 2024), are for simple grammar constraints, where highly-engineered parsers in performant languages like Rust or C++ can exploit the limited constraint expressiveness and systems engineering to evaluate the constraint for multiple possible next tokens simultaneously while navigating runtimes as low as 10 microseconds in the case of XGrammar (Dong et al., 2024). For further discussion of these libraries, see Cognetta et al. (2025) for a detailed overview.

ARS-LCD contributes a new way to integrate arbitrary black-box constraints that do more than check adherence to a simple grammar, for which such extensive performance engineering may not be feasible. In our experiments, constraints are written as short Python functions with minimal performance engineering; ARS-LCD lets them be used in the decoding loop without a severe hit to overall decoding speed. ARS-LCD thus supports both an increase in the complexity of what can be efficiently constrained at decode time as well as a usability benefit for constraint programmers, who can now afford milliseconds per constraint evaluation rather than being restricted to microseconds. We see many opportunities for interesting constraints at this time scale, including incremental type systems and simulators. Consider the Goal inference domain, for example, where a PDDL planner can be integrated into the decoding loop with ARS-LCD. In contrast, naive token masking for a model with a vocab size of $100{,}000$ would require an unusable $11$ms/tok $\times 100{,}000$tok $> 18$min/step.

## I.2 From Global Constraints to Local Constraints

The goal of controlled generation is to satisfy some sequence-level constraint. However, to enforce this constraint incrementally during generation, we need to formulate some local version of the constraint to check at each step. An important design question is how to implement this local constraint.

The optimal Boolean local constraint checks for membership in the prefix language of the sequence-level constraint, i.e., it checks that there exists at least one suffix that, when concatenated to the current prefix, satisfies the sequence-level constraint. In some cases, this prefix language can be evaluated quickly and precisely, e.g. in our pattern matching task, where we see performance near 100%.

But in many other cases, the exact calculation of this prefix language is intractable. Still, it is often relatively simple to implement fast local checks of validity, i.e., upper bounds on the optimal local constraint. For example, assume our goal is to generate Python code that

passes a series of tests. While it is not feasible to ask in the middle of the program whether it will eventually pass these tests, it is still simple to rule out many generations: the Python code should be syntactically valid, not reference undefined variables, use types correctly, and more generally execute without exception through the most recent complete statement, regardless of what it outputs.

When we talk about black-box constraint integration, this means that local constraint checkers may be arbitrary programs that map prefixes to Booleans. They need not adhere to some special restricted class of functions like regular grammars. This enables us to practically implement constraints like the one above, by writing a short program that actually calls the Python interpreter on generated code and checks if exceptions are raised. Existing frameworks for locally constrained decoding typically do not support this degree of freedom, instead requiring the local constraints to be expressed in restricted formal languages.

In general, it is up to the user to strike their own tradeoff between the local constraint's speed and precision. In general, more precise constraints will make AWRS-SMC more sample-efficient (better performance with fewer particles), whereas less precise constraints may mean that AWRS-SMC requires more particles to produce great posterior samples. But for all valid local versions of a constraint (i.e., all checkers that do not incorrectly rule out prefixes that could be completed to valid strings), the algorithm does correctly target the sequence-level conditional distribution, generating exact samples as the number of particles grows.

## J   Context-Sensitive Pattern Matching

In order to generate instances of pattern-matching specifications that exceed the expressiveness of standard FSM-based approaches, we executed the following procedure:

1. Prompt `claude-sonnet-3.7` to generate pattern-matching expressions to test a modern regex engine that use advanced features including lookaround, backreferences, conditionals, etc. This was repeated until generating 1503 candidates.
2. Drop duplicates.
3. Filter all proposed expressions that *do not* comply with the `regex` library (Barnett, 2014). This confirms these specifications are satisfiable.
4. Filter all proposed expressions that *do* comply with the `interegular` library (MegaIng, 2019). This confirms these specifications exceed the capacities of FSM-based approaches.
5. Confirm that the empty string "" is a valid prefix, but not complete. This satisfies our setting for benchmarking LM generation.

Following these steps, 402 valid test cases remained. The authors manually inspected a subset of the final dataset and found that it exploited a wide range of challenging specifications. Here are some examples:

- Repeating pattern of two characters in forward then reverse order:
  `^(\w)(\w)(?:\2\1)+$`

- Arbitrarily nested center embedding:
  `^(<<(?R)*>>|\w+)$`

- Conditional matching:
  `(\d{3})?(?(1)abc\1|xyz)`

- A mutually recursive arithmetic expression parser:
  `(?(DEFINE)(?<expr>(?&term)(?:[+\-](?&term))*)(?<term>(?&factor)(?:[*/](?&factor))*)`
  `(?<factor>\d+|\((?&expr)\)))^(?&expr)$`

- A simple JSON parser:
  `"^(?(DEFINE)(?<json>(?<obj>\{(?:(?&str):(?&val)(?:,(?&str):(?&val))*|)\})`
  `|(?<arr>\[(?:(?&val)(?:,(?&val))*|)\])|(?<str>""(?:[^""\\]|\\.)*"")|(?<val>(?&obj)`
  `|(?&arr)|(?&str)|true|false|null|-?\d+(?:\.\d+)?(?:[eE][+-]?\d+)?)))(?&json)$"`

## K   Experiment Details

This section provides details about the hyperparameters and hardware used in our experiments.

**Temperature & Max Tokens.**   All experiments were run with temp 1.0. For all methods, we set a maximum number of tokens threshold to prevent excessively long generation times. Thresholds are set for each domain depending on the typical length of valid sequences from that domain. The following thresholds were set: Pattern Matching (32), Molecular Synthesis (40), Goal Inference (100), Text-to-SQL (100), JSON (350).

**SMC Resampling.**   In our SMC-based methods (Twisted SMC, AWRS-SMC), a resampling step is triggered when the effective-sample size (ESS) of the particle beam falls below a set threshold. For the Pattern Matching, Text-to-SQL, and JSON domains, we use an ESS of $M/2$ for AWRS-SMC  and $M - 1$ for Twisted SMC, where $M$ is the number of particles. The Twisted SMC  threshold was chosen to trigger a resampling step as soon as a particle has been deemed invalid by the constraint. These domains used standard multinomial resampling. For the Goal Inference and Molecular Synthesis domains, we use an ESS of $M$, which triggers particle resampling at every step. In these cases, we used a stratified resampling scheme as in (Loula et al., 2025).

**Hardware.**   Text-to-SQL, Pattern Matching, JSON, and Molecular Synthesis experiments were run on a single L40S GPU, with the exception of any runs using Llama 3.3 (70B), which used 4 L40S GPUs. Goal Inference experiments were run on a single 40GB A100 GPU.

## L  Experiments with Language Models of Varying Size

| Method | Accuracy | Runtime (sec/ex) |
|---|---|---|
| Base LM | 0.159 (0.12, 0.20) | 0.04 (0.04, 0.05) |
| ARS-LCD | 0.953 (0.93, 0.97) | 0.07 (0.06, 0.08) |
| TM-LCD | 0.950 (0.93, 0.97) | 8.45 (6.17, 11.39) |
| Sample-Verify | 0.373 (0.33, 0.42) | 0.20 (0.19, 0.21) |
| Twisted SMC | 0.452 (0.41, 0.50) | 0.11 (0.10, 0.13) |
| AWRS-SMC | 0.974 (0.96, 0.99) | 0.29 (0.26, 0.33) |

(a) Llama 3.2 1B

| Method | Accuracy | Runtime (sec/ex) |
|---|---|---|
| Base LM | 0.570 (0.52, 0.62) | 0.10 (0.09, 0.11) |
| ARS-LCD | 0.993 (0.98, 1.00) | 0.13 (0.11, 0.14) |
| TM-LCD | 0.978 (0.96, 0.99) | 6.91 (5.68, 8.46) |
| Sample-Verify | 0.781 (0.74, 0.82) | 0.28 (0.26, 0.30) |
| Twisted SMC | 0.796 (0.76, 0.84) | 0.20 (0.19, 0.22) |
| AWRS-SMC | 0.990 (0.98, 1.00) | 0.36 (0.33, 0.40) |

(b) Llama 3.1 8B

| Method | Accuracy | Runtime (sec/ex) |
|---|---|---|
| Base LM | 0.818 (0.78, 0.86) | 0.22 (0.21, 0.24) |
| ARS-LCD | 0.993 (0.98, 1.00) | 0.25 (0.23, 0.28) |
| TM-LCD | 0.990 (0.98, 1.00) | 5.24 (4.50, 6.14) |
| Sample-Verify | 0.858 (0.82, 0.89) | 0.50 (0.46, 0.54) |
| Twisted SMC | 0.846 (0.81, 0.88) | 0.44 (0.41, 0.48) |
| AWRS-SMC | 0.995 (0.99, 1.00) | 0.50 (0.46, 0.55) |

(c) Llama 3.3 70B

Table 3: Comparison of method accuracy and runtime across language models of varying size on the Pattern Matching domain. Confidence intervals bootstrapped at the level of 95%. Runtime represents the average execution time (in seconds) across all instances in the dataset. Sample-Verify and Twisted SMC were run with $M = 10$ particles. AWRS-SMC was run with $M = 5$ particles. Llama 3.3 8B results are copied from Tab. 1e and repeated here for convenience. All models are instruct versions.

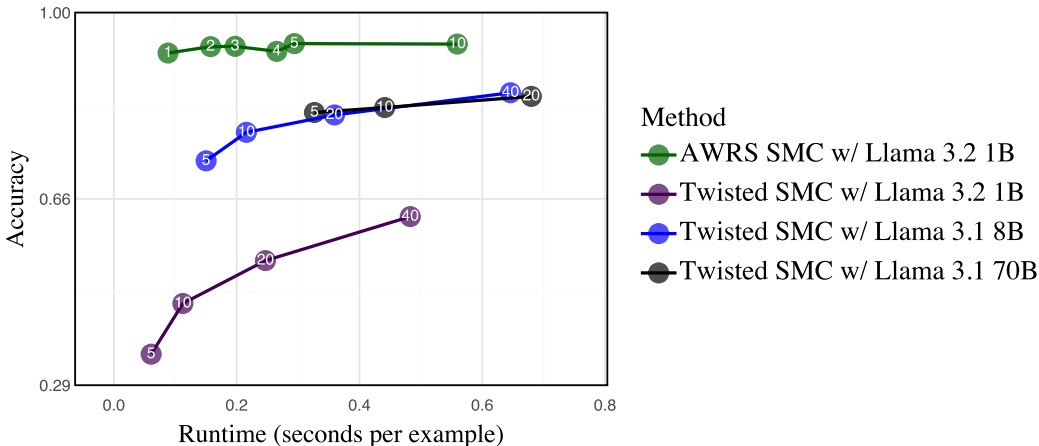

Figure L.1: Accuracy and runtime of AWRS-SMC and Twisted SMC with varying particle counts on the Pattern Matching domain, using LMs of different sizes. AWRS-SMC with a smaller LM achieves better performance than Twisted SMC with larger LMs at the same runtime cost. All models are instruct versions.

