# OpenReview forum: "Fast Controlled Generation from Language Models with Adaptive Weighted Rejection Sampling"
_colmweb.org/COLM/2025/Conference — COLM 2025_

### Official Review · Reviewer_VsDj · 2025-04-27

**Rating:** 8
**Confidence:** 4
**Ethics Flag:** 1

**Summary:**

This paper introduces the Adaptive Weighted Rejection Sampling (AWRS) algorithm to control the generation from Large Language Models under a constraint. It can bring significantly faster sampling compared to traditional token masking methods while maintaining the same distribution —interestingly it runs faster for better language models, which better capture the constraint. The authors characterize how the computational work scales with the divergence between the unconstrained and constrained language model. And experiments show improvements on five benchmarks.
The approach is theoretically sound –offering unbiased estimators that can correct the myopic behavior of local constraint enforcement when used with sequential Monte Carlo methods– and gives good empirical results —outperforming strong baselines in both accuracy and runtime efficiency. Furthermore, that AWRS' computational advantage increases with better models aligns well with the current trend.

**Reasons To Accept:**

This paper suggests a computational efficient algorithm (AWRS) that is both sound and versatile, showing good results with scaling benefits.
- computational efficiency: AWRS can significantly reduces constraint overhead, compared to typical Locally Constrained Decoding.
- theoretical soundness: the algorithm ensures an exact sampling from the desired distribution.
- versatility: the approach works with arbitrary constraints rather than being limited to specific classes.
- good experimental results across five domains: AWRS shows applicability in practical settings.
- scaling benefits: the method becomes more efficient with better LLMs well in line with with industry trends.

**Reasons To Reject:**

There are limitations to the approach, although I wouldn't say they are "reasons to reject" this work:
- stochastic runtime: runtime is stochastic and could vary significantly across instances, which can be an issue in production.
- limited performance gap in some domains: when the local constraint is already exact, e.g. Molecular Synthesis and Pattern Matching, improvements are less dramatic.
- computational overhead of multiple loops: while faster than token masking, the algorithm still requires multiple rejection loops to compute the unbiased Z estimator.

---

> ### Author Response · Authors · 2025-06-02
> **Response to VsDj**
>
> Thank you for your strong review and thoughtful questions! We address comments below.
>
> > stochastic runtime: runtime is stochastic and could vary significantly across instances, which can be an issue in production.
>
> Thanks for pointing this out. Yes, stochastic runtimes (particularly if the distribution of runtimes has long tails) can lead to issues in production. In Appendix H, we present a modified version of our algorithm that incorporates a user-specified upper bound on the number of attempts AWRS makes to generate a satisfying next token. This requires a slight change to the computation of proper weights for use within SMC, also presented in Appendix H. Note that, using this variant of our algorithm, if a single particle within SMC has found itself in a poor position, it will stop early with a weight of $0$, allowing resampling to cull the particle (and clone a more promising particle) without slowing down the rest of the algorithm. If a constraint is so difficult that the upper bound is hit frequently (across all particles in an SMC algorithm), the user may opt to increase their upper bound, dispatch to a larger model more capable of satisfying the constraint, or increase the number of particles in SMC to mitigate the issue. We will revise the paper to include a clearer discussion of these considerations.
>
> > limited performance gap in some domains: when the local constraint is already exact, e.g. Molecular Synthesis and Pattern Matching, improvements are less dramatic.
>
> As noted, in some settings the constraint is already quite exact, so additional corrections based on proper weighting may be unnecessary. We will make this point more clearly in our discussion of the results. In general, we would recommend that users try multiple settings for $M$, the number of particles in AWRS-SMC, and use the lowest setting that meets the functional requirements of their application. If $M=1$ is sufficient, then proper weighting has no effect, and the user can simply run ARS-LCD. However, in some domains, the proper weights really do help, and we see the ability to compute them without summing over the entire token vocabulary as a key benefit of our approach.
>
> > computational overhead of multiple loops: while faster than token masking, the algorithm still requires multiple rejection loops to compute the unbiased Z estimator.
>
> It is correct that the algorithm requires multiple loops to compute the unbiased $Z$ estimator. However, the overhead of this is rather minimal. In our experiments, we find that only $2$ total loops are sufficient, and these loops are not independent. Rather, the 2nd loop proceeds immediately after the first, which has already adaptively rejected some number of non-conforming tokens. The 2nd loop does not need to reject these again, and can even sample the same correct token that was already just sampled. Algorithm runtime is thus dominated by the first loop. For example, in the context-sensitive pattern matching domain, the second loop samples only $1$ token in $92$%, $85$%, and $72$% of runs for Llama 70B, 8B, and 1B, respectively. We will revise to discuss this point.

---

> > ### Comment · Reviewer_VsDj · 2025-06-07
> >
> > thanks for these answers to the limitations I had mentioned. This is indeed a "clear accept" for me.

---

### Official Review · Reviewer_cnre · 2025-05-11

**Rating:** 7
**Confidence:** 3
**Ethics Flag:** 1

**Summary:**

This paper introduces a novel sampling algorithm that could satisfy black box constraints and provides massive efficiency gains. The final algorithm is called Adaptive Weighted Rejection Sampling (AWRS). The core idea is behind the relationships of the magnitude of normalization constant and the number of tokens that were rejected before sampling the desired one. So authors found a way of how to oversample rejected tokens so that the WRS can be integrated into the sequential monte carlo (SMC). Further they come up with a way of how to use the probability mass of rejected tokens to correct the estimate of Z. This results in the algorithms that has expected lower runtime that they validate in the experiments.
During experiments they test their method as well as baselines on wide range of constraint decoding tasks such SQL and JSON generation. They show significantly faster runtime as well as higher quality of generations confirming the usefulness of the proposed algorithm.

**Questions To Authors:**

* What are black box constraints you mentioned in the intro? I noticed you wrote " compatible with any constraint" in your contribution bullet point. This is only about local constraint right? There are sequence level constraint that are hard to map into token level constraint i.e., those that can only be evaluated when sequence is completed. Would be great to have some clarification about this in the text.

**Reasons To Accept:**

* Algorithm is practically very useful and does not involve any significant computational overhead. I can see it being used in the standard inference engines such as VLLM or SGLang.
* LCD itself might be an exciting direction not only in the data generation or inference areas, but also in online training methods that are gaining lots of attention recently.

**Reasons To Reject:**

* There is no discussion of how this algorithm might be combined with truncation based sampling such as topp or \eta sampling. It would be great is authors would include some discussion about that.

---

> ### Author Response · Authors · 2025-06-02
> **Response to cnre (1/2)**
>
> Thank you for the review and for these thoughtful questions and comments! We answer all questions below.
>
> > There is no discussion of how this algorithm might be combined with truncation based sampling such as topp or \eta sampling. It would be great if the authors would include some discussion about that.
>
> Thanks for this suggestion! There are a few ways one might combine our method with truncation-based modifications to an LM’s sampling procedure.
>
> The most straightforward way is to think of truncation-based sampling as modifying the base LM distribution, on top of which our constrained decoding approach can then be applied. For example, in top-p sampling with $p=0.9$, we zero out the bottom $10$% of token probabilities and renormalize to obtain a modified next-token probability distribution. We can then apply our algorithm as though the base LM had generated this truncated next-token probability distribution: we adaptively rejection-sample from the truncated distribution, rather than from the full next-token distribution. One caveat is that the user must ensure that truncation does not remove all valid next tokens (according to the constraint) from the support.
>
> Another way to combine our methods with truncation-based sampling is to consider the truncation as part of the constraint: we write a new constraint function that accepts a token only if it passes the original constraint and is in the top $90$% of logits from the base model. If AWRS-SMC were applied with this constraint, the weights would correct not only for myopia from locally constrained decoding, but also for myopia from top-p decoding. As the number of particles grows, the algorithm’s output distribution would converge to the global posterior distribution that arises when the base LM is conditioned on the event that (1) the string satisfies the original constraint, and (2) each token is in the top $90$% of the LM’s predictions given the previous tokens. Note that, as above, the user must be careful to avoid the situation where there are no strings that satisfy this more stringent constraint.
>
> We will revise the paper to include discussion of these points!

---

> > ### Author Response · Authors · 2025-06-02
> > **Response to cnre (2/2)**
> >
> > > What are black box constraints you mentioned in the intro? I noticed you wrote " compatible with any constraint" in your contribution bullet point. This is only about local constraint right? There are sequence level constraint that are hard to map into token level constraint i.e., those that can only be evaluated when sequence is completed. Would be great to have some clarification about this in the text.
> >
> > Thanks for raising this important point. The goal of controlled generation is indeed to satisfy sequence-level constraints. But in order to enforce the constraint incrementally during generation, we need to formulate some local version of the constraint we care about. An important design question is how to implement this local constraint. The optimal Boolean local constraint checks for membership in the prefix language of the sequence-level constraint, i.e., it checks that there exists at least one suffix that, when concatenated to the current prefix, satisfies the sequence-level constraint. In some cases, this prefix language can be evaluated quickly and precisely, e.g. in our pattern matching task, where we see performance near $100$%. But in many other cases, the exact calculation of this prefix language can be intractable. Still, it is often relatively simple to implement fast local checks of validity, i.e., upper bounds on the optimal local constraint. For example, assume our goal is to generate Python code that passes a series of tests. While it is not feasible to ask in the middle of the program whether it will eventually pass these tests, it is still simple to rule out many generations: the Python code should be syntactically valid, not reference undefined variables, use types correctly, and more generally execute without exception through the most recent complete statement, regardless of what it outputs.
> >
> > __When we talk about black-box constraints, our point is that local constraint checkers may be arbitrary programs that map prefixes to Booleans.__ They need not adhere to some special restricted class of functions like regular expression matchers. This enables us to practically implement constraints like the one above, by writing a short program that actually calls the Python interpreter on generated code and checks if exceptions are raised. Existing frameworks for locally constrained decoding typically do not support this degree of freedom, instead requiring the local constraints to be expressed in restricted formal languages.
> >
> > In general, it is up to the user to strike their own tradeoff between the local constraint’s speed and precision. In general, more precise constraints will make AWRS-SMC more sample-efficient (better performance with fewer particles), whereas less precise constraints may mean that AWRS-SMC requires more particles to produce great posterior samples. But for all valid local versions of a constraint (i.e., all checkers that do not incorrectly rule out prefixes that could be completed to valid strings), the algorithm does correctly target the sequence-level conditional distribution, generating exact samples in the limit of infinite particles. We will clarify this discussion in the final version of the paper.

---

### Official Review · Reviewer_qHSZ · 2025-05-11

**Rating:** 7
**Confidence:** 4
**Ethics Flag:** 1

**Summary:**

The paper proposes a rejection sampling algorithm for constrained decoding and an unbiased estimator for important sampling that can be used to avoid greedy traps during decoding. They show that their algorithm is both performant and accurate.

**Questions To Authors:**

I think the people could benefit from some concrete walk-throughs of sequential Monte Carlo sampling as well as other sampling methods like the twisted method.

**Reasons To Accept:**

The algorithm is useful for a broad set of constrained decoding problems. The writing is fairly clear and easy to understand.

**Reasons To Reject:**

It is not clear to me that evaluating constraints is a major bottleneck in many constrained decoding scenarios, especially when LLM forward pass costs may dwarf constraint costs. That being said I can imagine more expensive constraint scenarios. It would be helpful to have comments in the paper about the relative costs of different common constraints.

It would also be nice to have an overview of actual constrained decoding implementations, as I can imagine that some constrained decoding methods may already be implemented in practice using adaptive rejection sampling.

I had difficulty locating proofs for some of the propositions. It would be nice to have links to proofs in the appendix in the main text.

---

> ### Author Response · Authors · 2025-06-02
> **Response to qHSZ (1/2)**
>
> Thank you for this review and for your thoughtful questions and suggestions! We address your comments below.
>
> > It is not clear to me that evaluating constraints is a major bottleneck in many constrained decoding scenarios, especially when LLM forward pass costs may dwarf constraint costs. That being said I can imagine more expensive constraint scenarios. It would be helpful to have comments in the paper about the relative costs of different common constraints.
>
> Thanks for this suggestion. The cost of constraint evaluation indeed varies dramatically depending on the complexity of the constraint. We will include the following table in the final version of the text, which shows the mean cost (ms/eval) of the constraints we test in this paper.
>
> |   |  Goal inference  |  Molecular synthesis  |  Pattern matching |  Text-to-SQL  |  JSON  |
> |---|---|---|---|---|---|
> |  Cost (ms/eval)  | 11.0 | 0.30  |  0.0054 |  6.5  |  0.57 |
>
> As can be seen, costs range from $<0.01$ms to $>10$ms, and yet all can still be integrated into the decoding process with ARS, incurring a comparatively low constant-factor overhead (mean $1.22\times$ sec/ex across our benchmarks). Contrast this with the full token-masking approach, which evaluates the constraint on all tokens at each step. This method is so costly that we could only tractably run the full token masking baseline for the context-sensitive pattern matching domain. Even there, the approach incurs $>50\times$ overhead. Without our algorithm, these more complex constraint evaluations can dwarf even LLM forward pass costs.
>
> Alternatively, most existing implementations of token masking are for simple grammar constraints, where highly-engineered parsers in performant languages like Rust or C++ can exploit the limited constraint expressiveness and systems engineering to evaluate the constraint for multiple possible next tokens simultaneously while navigating runtimes on the order of 10 microseconds (Dong et al., 2024). In these cases, your intuition that LLM evals dwarf constraint evaluation does hold, and token masking can be a highly effective technique.
>
> What our work contributes is a new way to integrate arbitrary “black-box” constraints that do more than check adherence to a simple grammar, for which such extensive performance engineering may not be feasible. Indeed, in our experiments, our constraints are written as short Python functions with minimal performance engineering; AWRS lets us use them in the decoding loop without a severe hit to overall decoding speed. An impact of this work is thus both an increase in the complexity of what can be efficiently constrained at decode time as well as a usability benefit for constraint programmers, who can now afford milliseconds per constraint evaluation rather than being restricted to microseconds. We see many opportunities for interesting constraints at this time scale, e.g., incremental type systems, simulators, and so on. Consider the Goal inference domain, for example, where a PDDL planner can be integrated into the decoding loop with ARS, whereas naive token masking would require $11$ms/tok * $256,000$tok > $45$min per timestep for a model with a vocab size of $256$k, such as Gemma.
>
> References:
> - Dong, Y., Ruan, C. F., Cai, Y., Lai, R., Xu, Z., Zhao, Y., & Chen, T. (2024). Xgrammar: Flexible and efficient structured generation engine for large language models. arXiv preprint arXiv:2411.15100.

---

> > ### Author Response · Authors · 2025-06-02
> > **Response to qHSZ (2/2)**
> >
> > > It would also be nice to have an overview of actual constrained decoding implementations, as I can imagine that some constrained decoding methods may already be implemented in practice using adaptive rejection sampling.
> >
> > Good point. At the time of writing this work, we checked major constrained decoding libraries, e.g., Outlines, Guidance, XGrammar, etc., and found that nearly all focused on limiting expressiveness to simple classes of grammars, which were pre-compiled to automata to efficiently materialize masks over the vocabulary at inference time. One exception was the CFG implementation in the Outlines library, which performed something related to adaptive rejection sampling—namely, they would sort the logits and check them in order, before returning the first compliant token. This has similar runtime benefits to ARS, but is deterministic, always yielding the same token across runs, and does not possess the probabilistic guarantees of adaptive rejection sampling, which faithfully samples exactly from the token masking distribution. Since submitting this work, we have become aware of one additional work, which uses an algorithm equivalent to unweighted adaptive rejection sampling for type-constrained code generation (Mündler et al., 2025). Their approach, like ARS LCD, yields performance benefits, but does not benefit from the weight correction methods of AWRS SMC. We have cited this new work in our draft and will add an expanded discussion comparing our approach to popular constrained decoding libraries.
> >
> > > I had difficulty locating proofs for some of the propositions. It would be nice to have links to proofs in the appendix in the main text.
> >
> > Thank you for raising this point. We will link to the matched appendix proof after each proposition in the main text.
> >
> > > I think the people could benefit from some concrete walk-throughs of sequential Monte Carlo sampling as well as other sampling methods like the twisted method.
> >
> > Thanks for mentioning this. We had originally deferred a light discussion of SMC to Appendix A for space concerns, but agree that a more thorough discussion would be valuable. We plan to add some more detailed walk-throughs the appendix and pull some intuitions as well as clearer references to the new appendix material into the main text within space constraints.
> >
> > References:
> > - Mündler, N., He, J., Wang, H., Sen, K., Song, D., & Vechev, M. (2025). Type-Constrained Code Generation with Language Models. arXiv preprint arXiv:2504.09246.

---

> > > ### Comment · Reviewer_qHSZ · 2025-06-03
> > >
> > > Thank you for your responses, I believe that these changes will strengthen the paper. I maintain my high score and recommendation for acceptance.

---

### Official Review · Reviewer_tTyg · 2025-05-14

**Rating:** 9
**Confidence:** 5
**Ethics Flag:** 1

**Summary:**

The authors present a rejection sampling approach to constrained decoding which leads to a more efficient implementation of decoding under arbitrary constraints. Interestingly, the computation required by the approach scales with the divergence between the constrained and unconstrained models, which is an appealing characteristic, because better models pay less decoding cost.

Especially for black box constraints, the proposed approach scales better than token masking. The authors first introduce weighted rejection sampling, which allows estimation of the normalizing constant Z, and then formulate Adaptive Weighted Rejection sampling, which allows adaptive rejection sampling while still allowing estimation of Z, leading to runtime benefits in practice. The paper is very clearly written, and the results outperform competitive benchmarks in both runtime and accuracy.

**Reasons To Accept:**

- performant constrained decoding approach which does not sacrifice too much speed
- broadly applicable and important topic
- clear exposition and thorough evaluation

**Reasons To Reject:**

- not clear whether implementation will be released, since this approach is engineering heavy, the lack of a reference implementation would be a significant bottleneck for future work

---

> ### Author Response · Authors · 2025-06-02
> **Response to tTyg**
>
> Thank you for this encouraging and thoughtful review! We address your comments below.
>
> > not clear whether implementation will be released, since this approach is engineering heavy, the lack of a reference implementation would be a significant bottleneck for future work
>
> Thanks for pointing out this was unclear. To clarify, in the final version of the paper, we plan to link to:
> 1) a lightweight, performant reference implementation that we have contributed to an existing open-source library for constrained generation.
> 2) a static snapshot of the code required to reproduce the experiments and analyses presented in the paper.
>
> The reference implementation is, in fact, already available, but we have withheld the link for anonymity during review.

---

### Author Response · Authors · 2025-06-02
**Response Summary**

Thank you for these thorough and thoughtful reviews!

In general, reviewers agreed that the paper made a significant contribution backed by strong theoretical and empirical evidence.

It was consistently noted that 1) the targeted topic is broad and important, 2) the approach presented in the paper is highly performant, practical, and in line with the field’s trends, 3) the evaluation was thorough with strong results across many domains, and 4) the exposition was clear and easy to follow.

Reviewers raised questions about 1) the availability of an open-source implementation, 2) increased discussion of constraint functions, including the relationship between global and local constraints as well as their associated costs, 3) a deeper walkthrough of AWRS SMC in direct comparison to the methods implemented in existing popular constrained decoding libraries, and 4) clarification for how to best utilize the algorithm, including a discussion of how to manage its stochastic runtime as well as its composition with other decoding strategies such as top-p.

In our individual reviewer responses, we have addressed each of these points, confirming that an open-source reference implementation will be linked to in the final version and adding clarifying discussion around the questions and points of interest raised by each reviewer. We look forward to continued discussion and providing any additional clarifications.

---

### Decision · Program_Chairs · 2025-07-08

**Decision:**

Accept

**Comment:**

The paper proposes a rejection sampling method for constrained decoding which allows them to handle arbitrary constraints in more flexible and efficient ways. The reviewers acknowledged the paper’s theoretical grounding, practical applicability, and clear exposition. The reviewers found the contribution significant. The reviewers raised concerns regarding the relationship between global and local constraints, and the need for comparisons to existing constrained decoding codes. The authors submitted a detailed response. The authors committed to publicly release their code, and provided detailed discussions on constraint function costs. They also explained further how their algorithm compares and relates to popular codes. The authors also provided additional explanations regarding the runtime, the combination with truncation-based sampling, and the definition of black-box constraints. The reviewers appreciated these clarifications and maintained their high scores. The area chair second the reviewers’ comments. The proposed method achieves impressive results, and its impact potential is high, provided the code is publicly released online. This is a clear accept, to consider for an oral presentation.